# Human cooperation in changing groups in a large-scale public goods game

Kasper Otten [1] ✉, Ulrich J. Frey [2], Vincent Buskens [1], Wojtek Przepiorka [1] & Naomi Ellemers [3]

How people cooperate to provide public goods is an important scientific question and relates to many societal problems. Previous research studied how people cooperate in stable groups in repeated or one-time-only encounters. However, most real-world public good problems occur in groups with a gradually changing composition due to old members leaving and new members arriving. How group changes are related to cooperation in public good provision is not well understood. To address this issue, we analyze a dataset from an online public goods game comprising approximately 1.5 million contribution decisions made by about 135 thousand players in about 11.3 thousand groups with about 234 thousand changes in group composition. We find that changes in group composition negatively relate to cooperation. Our results suggest that this is related to individuals contributing less in the role of newcomers than in the role of incumbents. During the process of moving from newcomer status to incumbent status, individuals cooperate more and more in line with incumbents.

Cooperation to provide public goods has been a central human activity throughout history. In our ancestral past, communal hunting, food sharing, and warfare produced public goods[1–3]. In contemporary life, important public goods include law enforcement, public education, social security, public transport, voting, and tackling climate change[4]. Despite the ubiquity of public goods, their provision is rarely trivial. Because every member of a group benefits from the public good, including those who do not contribute to its provision, individuals have an incentive to free-ride. However, if too many individuals free-ride, the public good is not provided and nobody benefits[5]. Public good provision is thus a social dilemma because individual and collective interests are not aligned. How groups can cooperate to overcome this social dilemma is a central scientific question that is still not fully answered.

Researchers have predominantly used economic game experiments to study the factors involved in cooperation in public good provision[6]. In typical public goods games, participants receive a monetary endowment and are asked to decide how much of it to keep for themselves and how much to contribute to a group project that also benefits other participants. Collective returns are maximized if everybody contributes their full endowment to the group project, but individual returns are maximized by not contributing anything, irrespective of what others do. More than a thousand studies have been conducted using public goods games[7], leading to many important insights into the motives (e.g., self-interest, reciprocity, fairness), institutions (sanctioning and reputation systems, social norms), and dynamics (e.g., conditional cooperation) related to human cooperation[8–12]. Many studies also suggest that cooperative behavior in public goods games is predictive of cooperative behavior in natural settings[13–18], although there is also some counter-evidence[19,20].

Public goods games are typically studied in one of two contexts: in repeated interactions among the same group members (also known as partner matching) and in interactions among members who change randomly after each interaction (also known as stranger matching)[9,21,22]. Research shows that contribution levels are considerably higher with partner matching than with stranger matching[23–27]. However, in real life, most public good problems occur in groups that lie between these two extremes, where group

---

[1]Utrecht University, Department of Sociology/ICS, Utrecht, the Netherlands. [2]Justus-Liebig-University Giessen, Faculty of Biology and Chemistry, Giessen, Germany. [3]Utrecht University, Department of Psychology, Utrecht, the Netherlands. ✉e-mail: k.d.otten@uu.nl

composition gradually changes over time due to old members leaving and new members arriving[28,29]. In such contexts, groups that produce public goods typically consist of a mix of incumbents and newcomers. Common examples are immigrants entering new countries, residents entering new neighborhoods, and employees entering new organizations and work teams. Despite their relevance for real-life groups, relations between ongoing changes in group composition and cooperation have not been studied systematically.

One reason for this may be high data requirements. Because groups are the units of analysis, sample sizes must be considerably larger than in experiments studying individual behavior. What is more, there are many ways in which group compositions can change over time, and the number of possibilities increases considerably with group size. This further amplifies the requirement for large sample sizes. In addition, groups need to be tracked for longer periods of time to observe compositional changes and their effects. So far, the few studies that relate changes in group composition to cooperation in public goods games have had to rely on relatively small sample sizes (between 100–300 participants, in groups of 2–6 members) and were only able to observe a limited set of group composition changes in a short time span[30–34].

There is a related literature on cooperation in dynamic networks. In dynamic networks, actors have some control over whom they interact with, allowing them to form and break ties with others based on others' cooperation decisions. Evolutionary models show that such strategic tie formation and dissolution can promote cooperation[35,36]. In particular, cooperation is expected to be higher if actors can frequently break with defectors and link with cooperative actors[37]. Behavioral experiments generally support these predictions; cooperation is higher in dynamic networks than in static networks and leads to clusters of cooperation[38–41]. However, this literature leaves largely unaddressed what happens in situations where individuals have little say in how the composition of their group changes and hence cannot easily break with defectors. For example, residents in a neighborhood typically do not get to choose who enters or leaves and employees in work organizations frequently have to accept with whom they have to collaborate based on the decision of their employers. What is more, exit costs are typically substantial in these situations, meaning that incumbents have little option to leave if they are dissatisfied with the newcomers. In sum, more research is needed on how group changes are related to cooperation when avoiding free-riders is not feasible.

In this paper, we analyze large-scale data from the multiplayer online game Ikariam, in which public goods games are deliberately built in by the designers and are central to players' success. The public goods games are played over a time span of multiple months in groups with a broad range of compositional changes. Moreover, options to leave the group or exclude free-riders are limited. The data are ideal to shed light on our two main research questions. First, what is the relationship between changes in group composition and cooperation in terms of contributions to a public good? Second, can this relationship be attributed to the contributions of the newcomers, the incumbents, or both? Two prior studies have shown that Ikariam players use contribution strategies that can be categorized as free-riding, conditional cooperation, and high cooperation[42,43], but how group changes relate to cooperation and whether newcomers and incumbents contribute differently has not yet been examined.

The theoretical answers to these questions can broadly be categorized into two opposing arguments. The first argument posits that newcomers will initially contribute less than incumbents because they lack a shared history with the incumbents. Newcomers may therefore have a lower concern for the group's welfare and a lower awareness of the norms prescribing contributions or a lower willingness to conform to them[22,29,44]. Newcomers are then expected to increase their contribution to the incumbents' level with more time spent in the group, as this increases the shared history they have with incumbents and allows them to get accustomed and socialized to the prevailing contribution norm in the group. The second argument posits that newcomers will initially contribute more than incumbents because they are under special scrutiny when entering the group and need to show their worth to the incumbents[45,46]. Hence, newcomers are expected to decrease their contribution to the incumbents' level with more time spent in the group, as their position will have been earned over time and the scrutiny decreases. Although both these arguments mainly suggest a role for newcomers' contributions in the relationship between changes in group composition and group cooperation, it is also possible that incumbents condition their contributions on changes in group composition. Indeed, some studies suggest that incumbents anticipate lower contributions by newcomers and other incumbents and will therefore reduce their contribution if newcomers enter[31,47].

In the Ikariam game, each player starts as the ruler of a town on an ancient Greek island with up to 16 other players on the island. On an island, individuals accumulate resources in real-time (i.e., the game continues after a player logs out) and have to make strategic decisions on how to use these resources. The resources can be invested in private goods, such as constructing and upgrading different types of buildings in one's town, e.g., a town hall, trading post, museum, or tavern. The resources can also be contributed to public goods. The main public good is a sawmill that provides wood. Wood is a crucial resource needed to develop one's town and hence to advance in the game. The sawmill is the main way for players to obtain wood. The rate at which individual players can extract wood from the sawmill depends on how many units of wood have been collectively contributed to the sawmill by all players on the island. This means that individuals who do not contribute nevertheless benefit from the contribution of other players on the island (non-excludability). The rate at which a player can extract wood from the sawmill does not depend on the rate at which other players extract wood (non-rivalry). The non-excludability and non-rivalry of the island's resource extraction make the game a pure public good analogous to public goods games[42,43]. More details on the public good dynamics in Ikariam are available in the Methods section.

An individual can enter additional groups by building a town on additional islands. Entering additional islands is an essential part of progressing in the game, as individuals eventually will need more resources than produced on their first island(s). When players newly join an island, they are able to extract resources from the island at a rate that depends on how much the incumbents of that island have contributed so far. Entering an island thus means entering a new group with a specific state of public good provision. An individual can become part of up to 12 groups, which means that an individual can become a newcomer several times. This also means that the same individual will sometimes be an incumbent in one group and a newcomer in another. Incumbents have no say in who enters their group and cannot exclude members.

We analyze longitudinal data on about 1.5 million contribution decisions of about 135 thousand players located in about 11.3 thousand groups. We examine the relationship between these contribution decisions and a total of about 234 thousand changes in group composition that occur over a time span of about a year. The results suggest a robust negative relationship between group changes and cooperation in public good provision. Newcomers contribute less to public goods than incumbents and thereby lower groups' average contributions. However, as newcomers spend more time in a group, they increase their contributions to the public good and contribute more in line with the group. That is, in the process of moving from newcomer to incumbent status, individuals' cooperation increases to the average level of the other group members.

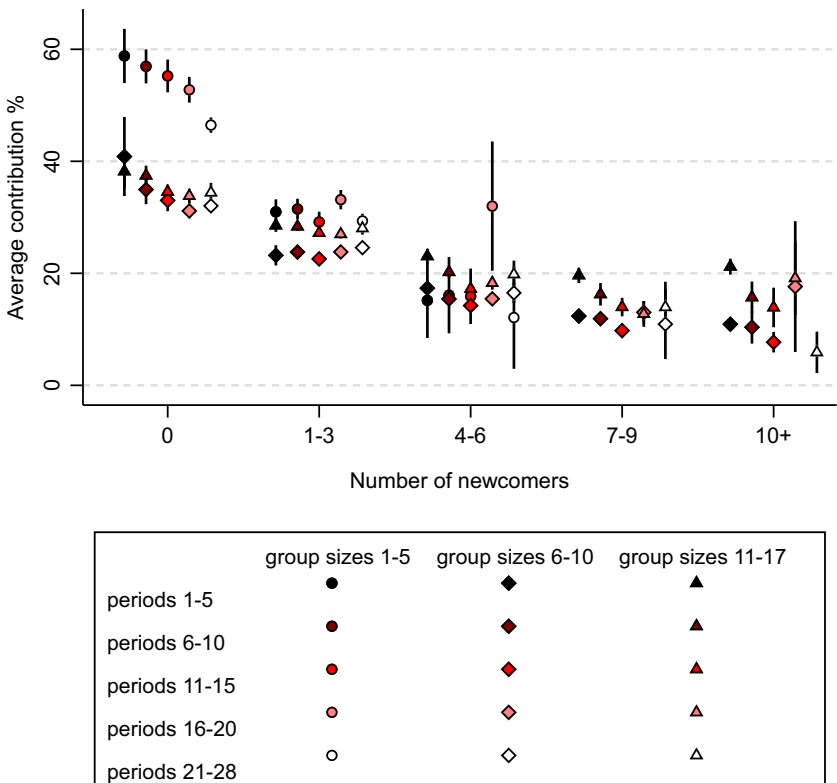

**Fig. 1 | Groups' average contribution percentages by the number of newcomers, group size, and time periods.** Data are presented as mean values and group cluster-robust 95% confidence intervals are provided via vertical spikes. The data are discretized in this figure for visualization purposes, the non-discretized analyses can be found in Table 1. The combination of a large number of newcomers and a small group size is not possible because a large number of newcomers implies a large group size. Therefore, no markers are shown for the combination of the group size category of 1–5 members and the upper two categories of the number of newcomers (7–9 and 10+ newcomers). Groups' average contribution percentage negatively relates to the number of newcomers. This holds for groups of different sizes, but more so for small groups (1–5 members) and also holds regardless of the time periods in which the newcomers enter. Results include 11,348 groups, with groups existing on average for 17–18 periods, giving a total number of observations of 199,530 group-period combinations. Source data are provided as a Source data file. Supplementary Code is provided to recreate the figure based on the Source data file.

## Results

The data are systematically structured in 28 biweekly intervals, i.e., we have one observation per two weeks per player-group combination. We refer to each biweekly interval as a time period, so we have 28 time periods (analogous to rounds of public goods games in lab experiments). Players' contributed resources are divided by their available resources to obtain their contribution percentage at each time period. We first examine the relationship between the average contribution percentage and the number of newcomers within each biweekly period. We regard an individual as a newcomer if the individual was not present in the group before the current period. We regard an individual as an incumbent if the individual was present in the group before the current period. Figure 1 shows how the group's average contribution percentage relates to the number of newcomers, group size, time periods, and the combination of these three variables. Note first that, for groups with zero newcomers, contribution patterns resemble those of aforementioned lab experiments with no newcomers in several aspects: (1) the initial contribution percentage lies between 40 and 60 percent, (2) the contribution percentage decreases over time, and (3), contribution percentages are higher in smaller groups. However, of particular interest to us is how the contribution percentage relates to the number of newcomers.

We see that a group's average contribution percentage negatively relates to the number of newcomers. The negative relationship is strongest when moving from no newcomers to a moderate number of newcomers (4–6) and is somewhat smaller when moving from a moderate to a large number of newcomers (+10). The contribution percentage decreases from about 50% to about 10% when moving from

the smallest to the highest number of newcomers. The negative relationship holds for groups of different sizes, although it appears stronger for small groups (1–5 members), and also holds regardless of the time periods in which the newcomers enter. Although the variables are discretized in Fig. 1 for visualization purposes, we do not discretize data in any of the statistical analyses and we find the same patterns there. All tests are two-tailed. Statistical tests using fixed effects regressions to account for between-group confounders reported in Table 1 confirm that the negative relationship between the number of newcomers and the average contribution percentage is significant and stronger for smaller groups. We also find that the negative relationship is significantly stronger at later time periods, although the size of this interaction is small. Out of the group size, time period, and number of newcomers, it is the number of newcomers that relates most strongly to the contribution percentage.

We next examine whether the relationship between the average contribution percentage and the number of newcomers also holds over the entire duration of the game instead of within time periods. Figure 2 shows the relationship between a group's average contribution percentage over all 28 time periods and the total number of newcomers that entered during this time. We once again see that there is a negative relationship between the number of newcomers and the average contribution percentage. Groups with very low total numbers of newcomers obtain contribution percentages of about 40–50% whereas groups with very large total numbers of newcomers obtain contribution percentages of about 10%. The negative relationship between a group's average contribution percentage and the total number of newcomers across all time periods is significant and

explains about 5 percent of the variation in contribution percentages (see Supplementary Note 1, Table S3). The bivariate correlation between the number of newcomers and the contribution percentage is −0.18 within periods and −0.23 across periods, which are regarded as small to moderate effect sizes in related research[48].

Since there is a maximum group size of 17 members, a total number of newcomers above 17 is only possible if there were also individuals leaving the group. The leavers thus make way for the newcomers. In fact, the number of newcomers and leavers are closely related in Ikariam; the total number of newcomers entering a group over all 28 periods correlates at 0.97 with the total number of leavers. Within periods, the correlation between the number of leavers and the

subsequent number of newcomers is 0.49, which gives us some room to examine whether the number of leavers also independently relates to the contribution percentage. In Figure S2 and Tables S15–16 of the Supplementary Information, we show that when including both the number of newcomers and the number of leavers as predictors of the contribution percentage, it is mostly the number of newcomers that is associated with lower contribution percentages. The negative relationship between the number of newcomers and the contribution percentage is robust to different operationalizations of the incumbent/newcomer and contribution variables, analyses excluding outliers, analyses per game server, analyses incorporating crossed fixed effects that control simultaneously for group and player characteristics, analyses controlling for the public good level, and other model specifications (see Supplementary Note 2, Tables S4–S14). We did not adjust *p*-values for multiple comparisons, but the relationships between group changes and cooperation would remain significant also when adjusting the cut-off *p*-values substantially downwards (e.g., when using $p < 0.001$ as the cut-off for significance instead of the conventional $p < 0.05$).

### Differences in contribution behavior between incumbents and newcomers

We next address what role incumbents and newcomers play in the negative relationship between the number of newcomers and incumbents, i.e., do newcomers contribute less than incumbents and/or do incumbents condition their contributions on the number of newcomers? To do so, we first look at the difference in contribution behavior between newcomers and incumbents. Since players in Ikariam are part of multiple groups and will occupy both the roles of incumbents and newcomers, we can perform a within-player analysis to assess whether being a newcomer is indeed associated with lower contributions. Such an analysis reduces the number of confounding factors due to between-player differences. Figure 3a shows that players contribute considerably less as newcomers than as incumbents. Whereas players contribute 32% on average as incumbents, they only contribute about 14% as newcomers. Even when we control simultaneously for player and group characteristics in a crossed fixed effects model, we find that incumbents contribute more than newcomers

**Table 1 | Regression model of average contribution percentages with group fixed effects**

|  | Model 1 | Model 2 |
| --- | --- | --- |
| Number of newcomers | −2.72*** | −4.87*** |
|  | (0.04) | (0.06) |
| Group size | −0.32*** | 0.04 |
|  | (0.03) | (0.03) |
| Period | −0.27*** | −0.33*** |
|  | (0.01) | (0.01) |
| Number of newcomers × group size |  | 0.48*** |
|  |  | (0.01) |
| Number of newcomers × period |  | −0.05*** |
|  |  | (0.01) |
| Intercept | 33.75*** | 33.75*** |
|  | (0.07) | (0.07) |
| R² (overall) | 0.04 | 0.05 |
| Rho | 0.58 | 0.58 |

*$p < 0.05$, **$p < 0.01$, ***$p < 0.001$. Linear regression with group fixed effects to account for repeated measures within groups. Coefficients of independent variables and intercept are marginal effects. Standard errors are provided in parentheses. Statistical significance is calculated using two-sided *t*-tests. Results include 11,348 groups, with groups existing on average for 17–18 periods, giving a total number of observations of 199,530 group-period combinations. All variables are entered without discretization.

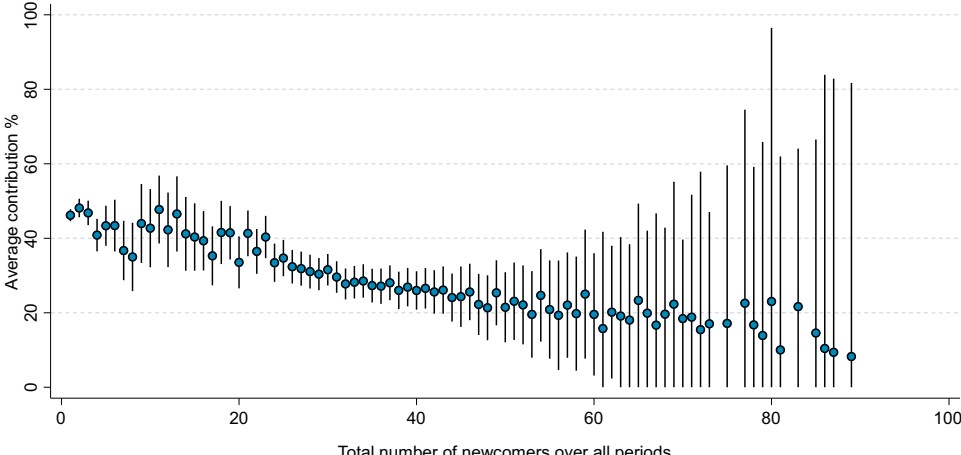

**Fig. 2 | Average contribution percentages by the total number of newcomers over all periods.** Data are presented as mean values and 95% confidence intervals are provided via vertical spikes. In contrast to Fig. 1, we look at a group's average contribution percentage over all 28 periods, which means there is only 1 observation per group. Results include 11,353 groups. At any given period, a group can only consist of up to 17 players and hence only up to 17 newcomers can enter. However, groups may also experience incumbents leaving in some periods, opening up new spaces for newcomers. So the total number of newcomers across all time periods can be higher than 17. Because there are few groups with a very high total number of newcomers, the contribution percentages for these groups have larger confidence intervals. We cut off confidence intervals below 0 because contribution percentages below 0 are not possible. We see a negative relationship between the group's average contribution percentage across all time periods and the total number of newcomers that have entered during this time. In Table S3 of the Supplementary Information, we show that this relationship is significant, also when controlling for the group's average group size and time period. Source data are provided as a Source data file. Supplementary Code is provided to recreate the figure based on the Source data file.

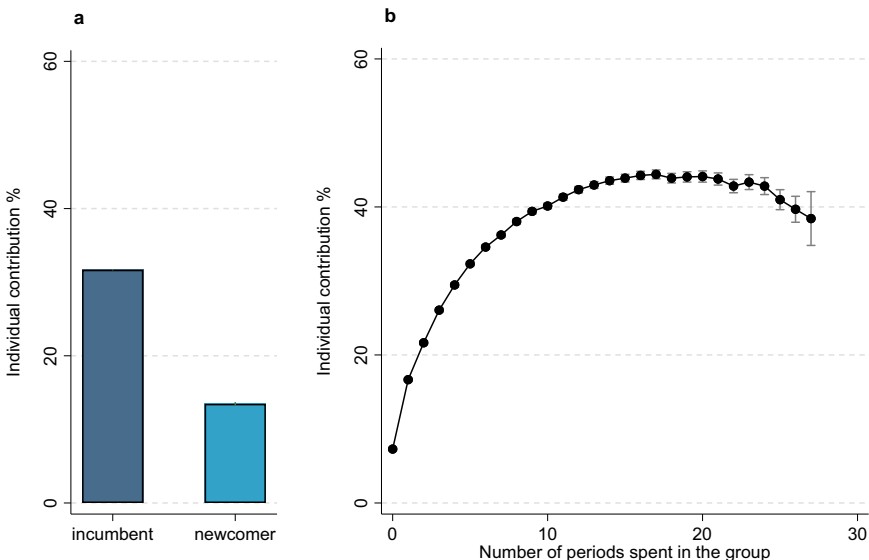

**Fig. 3 | Individual contribution percentage by newcomer status and time spent in the group.** For newcomer status (**a**), results are based on a player fixed effects regression with the individual's contribution percentage as the dependent variable and a factor on whether the individual is a newcomer or incumbent as the independent variable. We control for the time period and group size. For time spent in the group (**b**), results are based on a player fixed effects regression with the individual's contribution percentage as the dependent variable and a factor on the

number of periods spent in the group as the independent variable. We control for the time period and group size. Data are presented as mean values and 95% confidence intervals are included via vertical spikes. Results for both panels include 1,572,734 contribution decisions made by 134,631 players. Source data are provided as a Source data file. Supplementary Code is provided to recreate the figure based on the Source data file.

**Table 2 | Regression model of incumbent contribution percentages with group fixed effects**

|                       | Incumbents' contribution |
|-----------------------|--------------------------|
| Number of newcomers   | −0.33***                 |
|                       | (0.05)                   |
| Group size            | −0.72***                 |
|                       | (0.03)                   |
| Period                | −0.43***                 |
|                       | (0.01)                   |
| Intercept             | 37.35***                 |
|                       | (0.08)                   |
| $R^2$ (overall)       | 0.03                     |
| Rho                   | 0.61                     |
| N                     | 187,758                  |

*$p < 0.05$, **$p < 0.01$, ***$p < 0.001$. Linear regression with group fixed effects to account for repeated measures within groups. Coefficients are marginal effects with standard errors provided in parentheses. Statistical significance is calculated using two-sided $t$-tests. Results include 10,940 groups (groups with no incumbents are excluded), with these groups existing on average for ~17 periods, giving a total number of observations of 187,758 group-period combinations. All variables are entered without discretization.

(Supplementary Note 2, Table S13). Figure 3b shows that players increase their contribution percentage in the process of moving from newcomer to incumbent status. Whereas the contribution percentage is low when first entering a group, it increases steadily with more periods spent in the group. After ten periods in the group, the contribution percentage is 40% and remains mostly stable afterward. Thus, in the progress of moving from newcomer to incumbent status, players increase their contribution percentage.

That players contribute considerably less as newcomers than as incumbents suggests that newcomers' contribution behaviors play a role in the negative relationship between the number of newcomers and the average contribution percentage. We can further assess the role of newcomers in this negative relationship by examining whether

the relationship remains when subtracting newcomers' contributions in calculating the group-average contribution. That is, we examine the relationship between the number of newcomers and the incumbents' contribution percentage. Table 2 shows that the negative relationship is indeed strongly reduced. Whereas originally each additional newcomer was associated with a 2.72 lower average contribution percentage (Model 1 in Table 1), each additional newcomer is only associated with a 0.33 lower contribution percentage among incumbents (Table 2). Although this association is still significant, it is only a small fraction of its original size, which suggests that most of the negative relationship can be linked to the newcomers' (lack of) contributions.

### Mechanisms behind the newcomer-incumbent difference

The finding that players contribute less as newcomers than as incumbents but do contribute more the more time they spend in the group is in line with the theoretical mechanism that newcomers need time to become accustomed to and socialized with the prevailing contribution norm in the group. To further delve into this norm-based mechanism, we examine the extent to which a player's contribution percentage relates to the group's average contribution percentage (excluding the player's own contribution percentage) and how this develops with time spent in the group. An incumbent's contribution percentage correlates at 0.21 with the group's average contribution percentage. A newcomer's contribution percentage correlates at 0.13 with the group's average contribution percentage, significantly lower than the correlation for incumbents (Supplementary Note 3, Table S17). This suggests that newcomers indeed contribute less in accordance with the group contribution norm than incumbents. What is more, the relationship between a player's contribution percentage and the group's contribution percentage increases with time spent in the group (Supplementary Note 3, Table S17 and Figure S4). That is, as players spend more time in the group, they contribute more in line with the group contribution norm. This is further in line with the norm-based mechanism specifying that newcomers need time to get accustomed to the prevailing contribution norm in the group.

We find little evidence that having a shared history with the incumbents in and of itself (i.e., without socialization to the group's contribution norm) is related to higher contributions. To assess this, we examine whether newcomers who already know the incumbents from their other groups contribute more than newcomers who do not know the incumbents from other groups. Although newcomers' contribution percentage is slightly higher if they know more incumbents from prior groups, the effect size is very small (0.7 percentage points, Supplementary Note 3, Table S18). The difference between newcomers and incumbents is also present when comparing players who are part of only one group and hence do not have to share their attention across groups (Supplementary Note 3, Table S19). We also find little evidence that newcomers use the contribution norm of their other groups to inform their contribution decision in new groups (Supplementary Note 3, Table S20). Instead, newcomers seem to start with low contribution percentages in their new groups and over time contribute more in line with the group's norm.

We further examine to what extent the difference between newcomers' and incumbents' contribution percentages is related to inequality between them in how much they can contribute to, and benefit from, the public good. The role of inequality in resources between incumbents and newcomers is largely prevented by examining not absolute contributions but instead the percentage of one's resources contributed. Newcomers can thus always achieve as high contribution percentages as incumbents, and not doing so is a choice. Still, having more resources might motivate one to contribute higher percentages, as having more resources increases the efficacy of one's contribution. Similarly, lower benefits might also lead newcomers to choose to contribute lower percentages.

We examine the role of inequality in resources and benefits in two ways. First, incumbents generally have higher-level private goods than newcomers. A player's private good level on an island is captured by the player's town hall level. Every increase in a player's town hall level increases the maximum number of citizens allowed in the player's town. Because citizens produce resources, an increase in the maximum number of citizens generally means an increase in resources. Hence, a higher town hall level means more resources. Because incumbents generally have higher private good (town hall) levels than newcomers, they have more resources to contribute to the public good and also benefit more from contributing because they are more in need of strong public goods to support their higher private good levels. Without controlling for private good level, we estimated the incumbents' contribution to be on average 18 percentage points higher than the newcomers' contribution (see Fig. 3A). Controlling for the private good level decreases the difference in the contribution percentage between newcomers and incumbents from 18% to 9% (Supplementary Note 3, Table S21). Hence, while the newcomer-incumbent difference is halved when controlling for private good levels, it remains substantial and significant.

A second way to examine the role of inequality in the efficacy and benefits of contributing is by examining the difference between newcomers' and incumbents' contribution percentages depending on the public good level. If public good levels are low, it takes little resources to increase the public good level and the returns of increasing the public good level are high (see also Supplementary Information, Table S2). In this situation, both newcomers and incumbents are in a position to effectively contribute to the public good and benefit from doing so. If public good levels are high, it takes more resources to increase the public good level and returns are lower. In this situation, incumbents are in a better position than newcomers to effectively contribute due to their higher amount of resources and benefit from doing so. Hence, if inequality in benefits and efficacy of contributing matters, we would expect the difference in contribution percentages between newcomers and incumbents to be lower with lower public good levels. Indeed, we find that the difference in the contribution

percentage is lower with lower public good levels (Supplementary Note 3, Table S22). Whereas the average newcomer-incumbent difference in contribution percentage is about 18% (Fig. 3A), the difference is about 9% at the lowest public good level (Supplementary Note 3, Table S22). Again, this shows that the newcomer-incumbent difference is smaller when incentives to contribute are similar for newcomers and incumbents, but also that it remains substantial and significant.

## Discussion

Using large-scale data from a multiplayer online game that incorporates public good dilemmas, we find that changes in group composition relate negatively to contributions to the public good. This negative relationship holds both when looking at short-term group changes and when looking at the total number of group changes across the entire observed duration of the game. Although incumbents slightly decrease their contributions if newcomers enter, the negative relationship between group changes and contributions is linked mostly to players who enter the group as newcomers. That is, newcomers contribute considerably less than incumbents and thereby lower groups' average contribution percentages. However, as players spend more time in their new group, they increase their contributions to the public good.

The results are consistent with group socialization models suggesting that newcomers gradually increase their tendency to act in line with the group's welfare and norms when making the transition from outsider to insider[49,50]. Indeed, individuals do not only contribute more if they are longer in the group, their contribution behavior also starts to resemble that of their group members more. Hence, individuals act more in line with the prevailing contribution norm as their time in the group increases. This norm-based mechanism seems to be more important than shared history explanations; newcomers who already know the incumbents from other groups hardly contribute more than newcomers who do not know the incumbents. Our findings also suggest that part of the newcomer-incumbent difference in contribution behavior is related to newcomers being in a disadvantaged position in terms of how effectively they can contribute to the public good and benefit from it.

The dynamic networks literature suggests that group changes can promote cooperation when they allow individuals to create ties with cooperative actors and break or avoid ties with uncooperative actors. However, not all situations allow individuals to determine how the composition of their group changes. For example, residents usually do not get to select who enters or leaves their neighborhood, and employees in many work organizations often do not get to form their own teams. Because exit costs are also typically substantial in these situations, incumbents cannot easily leave their group if they are dissatisfied with the newcomers. Similarly, incumbents in Ikariam have no say in who enters their group and, since exit costs are high, leaving the group is usually not an option. Hence, in contrast to most of the dynamic networks literature, strategically linking with cooperative actors and breaking with defectors is not a solution to cooperation in our study context. This may explain why we find a negative relationship between group changes and cooperation instead of a positive.

Our finding that newcomers' contributions are initially lower than those of incumbents but do increase over time resonates with field studies on newcomer contributions. Studies suggest that residents are more likely to volunteer at community events if they are longer part of the community[51], that workers' output in organizations is higher with higher tenure[52], and that immigrants contribute more to charitable organizations with more time spent in the country[53]. Typically, these studies only observe individuals at one point in time, so changes over time within individuals as they switch roles are not accounted for. In our study, the same individuals take both the roles of newcomer and incumbent, allowing us to compare within individuals how contribution

behavior changes depending on one's role in the group. This is important for current debates on newcomer contributions, in which a common argument is that group changes have negative effects because newcomers have lower dispositions to contribute and are therefore expected to contribute structurally different from incumbents[54]. For example, immigrants are sometimes said to contribute less than native populations because they come from different nations with different levels of public good provision[29]. Our study allows us to rule out individual dispositions and assess whether individuals condition their contributions on their role in the group. That we find a contribution difference between newcomers and incumbents while ruling out individual dispositions suggests that one's role in the group is a meaningful element in the difference between newcomers' and incumbents' contributions. Individuals' contribution percentages show a clear increase when roles switch from newcomer to incumbent.

What is more, the initially lower contributions by newcomers do not seem to be related to a motivation to take advantage of the incumbents, as is sometimes feared[29]. Rather, newcomers initially have lower benefits and efficacy of contributing to public goods. With more time spent in the group, newcomers' benefits and efficacy of contributing increase and they increase their contributions to the incumbents' level. This finding on the role of inequality in benefits/efficacy of contributing is in line with prior theory[55,56] and empirical research[57,58] suggesting that inequality can hamper cooperation. Similar processes may play a role in explaining immigrant contributions to public goods. For example, cross-sectional research suggests that part of immigrants' lower contributions to public goods may be attributed to their lower education and income level[53]. If immigrants have higher education and income levels, they are in a better position to contribute to public goods and also do so. The link between newcomers' disadvantaged position and their lower contribution implies that, rather than marginalizing newcomers' contributions or avoiding group changes, it is better to give newcomers the time to adjust and put them in a better position to effectively contribute to public goods[59–61].

Similarly, our finding of a negative relationship between changes in group composition and contributions to the public good should not be interpreted as evidence that change is bad. Changes in group compositions are unavoidable; incumbents will leave their groups at some point. Initial low contributions by a newcomer are better than no contributions at all if incumbents are not replaced. This becomes especially apparent if one considers the long-term contribution potential of newcomers whenever they are given the time to transition to an incumbent role. Our results simply suggest that to understand a group's current contribution to its public goods, it is informative to know its composition in terms of newcomers and incumbents.

Compared to typical research using public good games, our sample is broader and more heterogeneous. The inclusion of players from Germany, the United Kingdom, France, Greece, and Turkey means our sample goes slightly beyond typical WEIRD samples (Western, Educated, Industrialized, Rich, and Democratic)[62]. A survey reporting the average age of Ikariam players to be around 31 years[42] suggests that our sample more closely reflects the global median age than most other public good game studies which predominantly recruit younger university undergraduates[63]. Whereas social dilemma studies are typically somewhat overrepresented by women[7], Ikariam is largely overrepresented by men (~80% men) as is common for computer games. We do not have access to data on the income or education of Ikariam players, so cannot establish representativeness in these aspects. Altogether, our sample presents an improvement in terms of representativeness in some areas (e.g., global coverage and age), but still has limited representativeness in other areas (e.g., sex).

Virtual worlds such as Ikariam present an exciting and growing opportunity to study cooperation in context-rich settings over longer periods of time and broader ranges of group compositions. These virtual worlds offer opportunities to unobtrusively track behavior of all individuals in an entire population within a constrained and well-understood environment. Lab experiments are still needed to draw causal inferences and field studies to bring external validity. However, virtual worlds provide an insightful addition to these more traditional research methods to together provide a fuller understanding of human cooperation.

## Methods

### Game context

Ikariam is a free online browser-based strategy game that has been played by more than 50 million individuals so far. Results from a survey suggest that about 80 percent of the players are male, with an average age of 31 years[42]. The game is financed by some players paying real money to unlock in-game advantages such as obtaining more resources. Although players do not have a monetary incentive to act selfishly or cooperatively in the game, there are real incentives in terms of time investment and game progression. Because the game is played in real-time (i.e., the game continues after a player logs out), players typically have to log in multiple times per day to be sure that their towns keep running well and to respond to unforeseen events such as running out of resources. How quickly people progress in the game during this time is dependent, among other factors, on how selfishly or cooperatively they behave. Competition via leaderboards further incentivizes individuals to perform well.

The game is set in an ancient Greek archipelago and each island is regarded as a separate group tasked with producing public goods. The composition of the island will change over time as newcomers enter the island and incumbents leave the island, similar to how the composition of real-life groups producing public goods, such as countries, neighborhoods, and organizations, changes over time. A survey of Ikariam players confirms that they are well aware that contributing to the island is a cooperative act and that there are incentives to free-ride on others' contributions[42]. The presence of the public goods problem in Ikariam is further indicated by the language used in the community. Ikariam players have a specific term for free-riding, namely leeching, in their online community pages. They can read about leeching on the wiki[64], and can even find user-built tools to detect leechers (free-riders) in their group based on different contribution rules[65].

The context of public good provision in Ikariam falls between the constrained setting of the lab and the unconstrained setting of the field. Compared to the lab, public good provision in Ikariam is context-rich, long-term, free of observer bias from experimenters' presence, and observed among a more diverse pool of individuals. These features make the context of Ikariam arguably more similar to field settings of public good provision[19,62,66]. There is a growing body of research that examines whether cooperation patterns found in the lab also translate to field settings of public good provision[67–69]. For example, the peer-production of Wikipedia has been analyzed and compared to public good provision in lab experiments[17,70,71]. Such studies of public good provision in the field generally improve external validity, while lab studies remain important to provide a constrained environment that reduces the possibility of confounding variables. Compared to the complex environment of public good provision in the field, Ikariam consists of a highly standardized environment of which every aspect is recorded. Thus, in contrast to most field studies, we have information on the entire context in which public good provision takes place. All groups face the exact same public good problem and contribution behavior is therefore directly comparable across groups and individuals. Altogether, Ikariam provides a middle ground between a constrained lab environment and an unconstrained field setting.

### Public good dynamics

There are two public goods per island, a sawmill producing wood and an island-specific good producing wine, marble, crystal glass, or sulfur.

Both public goods work exactly the same, it is only the produced good that differs. For simplicity, we use the sawmill as an example when explaining the public good mechanism. In doing so, we refer to several existing variants of public goods games that bear resemblance to public good provision in Ikariam. Four specific characteristics of the public good in Ikariam are (a) it provides returns in real-time, (b) the returns increase step-wise by the total contribution of all group members, (c) the public good is durable, and (d) individuals differ in how much they can absolutely contribute.

The sawmill produces wood for each group member at a certain rate per hour, with the rate depending on how much wood has been contributed to the sawmill in total by all group members (players on the island). At the start of the game, when nobody has contributed yet, the sawmill produces 30 units of wood per hour for each group member. Hence, over 10 hours, a player would receive 300 units of wood. The hourly production rate can be increased if group members contribute wood to the public good. However, wood contributed to improve the sawmill cannot be used to develop one's town, giving an incentive to free-ride on the contribution of others. The public good increases step-wise as a function of the total contributions made to it by all group members. For example, the rate of 30 units of wood per hour can be increased to 38 units of wood per hour if all members combined contribute 394 units of wood to the public good. The public good is then said to have increased from level 1 (return of 30 units of wood per hour) to level 2 (return of 38 units of wood per hour). In total, there are 50 steps of improvement in succession. The step-wise increase in the benefits of a public good after total contributions surpass a threshold is commonly studied in lab experiments with step-level public goods games[72,73]. We provide the thresholds and the step-returns associated with these thresholds in the Supplementary Information (Supplementary Methods 2, Table S2). The continuous-time flow of benefits from the public good is akin to continuous-time public goods games[74,75].

The public good is durable: if a certain production rate per hour has been reached, it will never drop back to a lower rate. Relatedly, contributions are cumulative and not rebated. For example, if an individual contributes 500 out of 1000 units of wood required to move to the next public good level, the 500 remains in the sawmill even though the public good level (and hence the hourly production rate of wood) is unaffected. Only if another 500 units of wood are contributed to surpass the threshold, the hourly production rate of wood increases. See for prior empirical work on durable public goods games[76,77]. Finally, individuals that have accumulated more resources in the game can also contribute more. This is similar to the dynamic public goods game, where the wealth accumulated in prior rounds determines the endowment that can be contributed to the public good in subsequent rounds[78]. To compare contributions across individuals with different endowments, we examine not the individuals' absolute wood contributed, but instead the percentage of wood contributed out of the total wood that they had available on the island at the time of measurement.

Players can see the contributions of other players on the island at any time (an example is provided in Supplementary Methods 1, Table S1). This observability of the contributions of all members is an important element that allows for cooperation norms to be at play via reciprocity, where individuals can condition their contributions on their group members' contributions[79,80]. That contributions on each island are observable to all members allows us to examine whether players indeed contribute more when their group members also contribute more. A contribution of our study is that it also allows us to examine to what extent a player's tendency to contribute in line with the group differs between newcomers and incumbents, and whether newcomers contribute more in line with the group as they spend more time in the group.

Once an individual has entered a new group, it is generally not possible to leave, with two exceptions. The first is if an individual quits the game altogether. The second is if an individual spends real money to be able to move their town from one group to another, which is very rare. An individual can choose any group to enter, as long as the group has not reached the maximum group size of 17 yet. Incumbents thus have no say in who enters their group and cannot exclude members. When choosing which group to enter next, individuals have information on the island-specific resource that is produced, the location and size of the group, the incumbents in the group, and the current level of the public goods in the group. Generally, individuals will prefer groups that produce the island-specific resource that individuals are most in need of, groups that are located close to their current group(s), and groups with high public good levels.

There is the option to attack other players in Ikariam. Attacks are not publicly seen by others and can serve multiple purposes. For example, they can be used to take away resources from other players or to sanction those who do not contribute enough. However, attacks are very costly to both the attacker and the player defending the attack because both attacking and defending require armies that take up large amounts of resources. Attacks therefore only occur infrequently and are not central to the gameplay of most players. Since we do not have data on attacks and because they are not central to the gameplay, we do not focus on attacks in this study.

## Data collection and analysis

The data were collected and provided by Gameforge, the creator of the game. Players of the Ikariam game provide consent for (third-party) analyses of their non-personal data when signing up for the game. We did not have any access to personal data and obtained ethical approval for the study protocol from the Faculty Ethics Review Board of the Faculty of Social and Behavioural Sciences of Utrecht University. We have data from five servers, each from a different country: Germany, the United Kingdom, France, Greece, and Turkey. Each server contains a fixed number of 5351 islands, but the number of players differs per server. Data collection is identical and synchronous for each country. Given that there are no large differences by country[43], we pool the data across countries. The data is structured in 28 biweekly snapshots between April 2013 and February 2014 (the snapshots are biweekly on average; they start out weekly in April, get biweekly in August, and still later it is a 4-week interval). The first snapshot coincides with the start of new game servers, so we begin our observation at the actual beginning of a game. The data were analysed using Stata MP 15.1.

Since group size ranges from 1 to 17 players per group, and each player can either be a newcomer or an incumbent at each time period, we can observe many different combinations of the number of incumbents and newcomers. This allows giving a comprehensive answer to our first research question on how changes in group composition relate to contributions to the public good. We take advantage of the longitudinal nature of the data by examining this relationship within groups, which reduces potential confounding by between-group differences.

Furthermore, because players in Ikariam are part of multiple groups and experience both roles of newcomer and incumbent, we can examine differences in contribution behavior between newcomers and incumbents within players. By analyzing whether the same player contributes differently depending on whether the player is an incumbent or newcomer in the group, we can exclude selection effects based on different personal characteristics and assess whether just one's role in the group already relates to contribution behavior. This allows us to rule out individual disposition when answering the second research question concerning differences in contribution behavior between newcomers and incumbents, which is typically not possible in related prior research. For example, when finding differences in contributions to national public goods between migrants and native populations, it is typically not possible to pinpoint whether these differences arise from

the role that one has in the group (newcomer vs incumbent) versus selection effects (migrants having different individual dispositions than native populations)[81].

Recall that there are two public goods that individuals can contribute to in each group (the sawmill and the island-specific good). Resources that an individual contributes to one of the two public goods cannot be contributed to the other public good. The contributions to the two public goods are added up and divided by the total resources available at the time of measurement to obtain an individual's contribution percentage. Analyses that examine each public good separately are provided in the Supplementary Information and show no substantial differences between the two (Supplementary Note 2, Tables S7–9).

Because individuals can move resources between their groups, it can happen that they contribute more to the public good of a group than the total resources they had available in that group, i.e., individuals can end up with contribution percentages above 100 percent. Likewise, because we only have snapshots of an individual's available resources instead of a continuous-time overview of an individual's resources, it is possible that an individual had more (or fewer) resources available than we see at the snapshot, which can also lead to contribution percentages above 100 percent. This happens in 6.5% of our analyzed cases. In the Supplementary Information (Table S11), we show that the negative relationship between the contribution percentage and the number of newcomers remains significant when leaving out these cases or setting them to 100 percent.

### Reporting summary

Further information on research design is available in the Nature Research Reporting Summary linked to this article.

## Data availability

The raw data is the property of Gameforge, the producer of the game. Gameforge allowed the authors to use the raw data for academic purposes, but did not allow the raw data to be openly shared. Requests for access to the raw data can be directed to Ulrich Frey, who can be contacted at uf@ulrichfrey.eu. We provide an aggregated dataset at https://doi.org/10.17605/OSF.IO/3WYE9. Source data are provided with this paper.

## Code availability

We provide Supplementary Code to reproduce the figures of the paper based on the Source data file.

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

## Acknowledgements

We are very grateful to Gameforge for providing the data. This study is part of the research program Sustainable Cooperation – Roadmaps to Resilient Societies (SCOOP). The authors are grateful to the Netherlands Organization for Scientific Research (NWO) and the Dutch Ministry of Education, Culture and Science (OCW) for generously funding this research in the context of its 2017 Gravitation Program (grant number 024.003.025).

## Author contributions

K.O. carried out the investigation, wrote the original draft, and made the visualizations. K.O. and U.J.F. conceptualized the research and conducted the formal analysis. K.O., U.J.F., V.B., W.P., and N.E. were responsible for the methodology and review & editing of the writing.

## Competing interests

The authors declare no competing interests.

## Additional information

**Correspondence and requests** for materials should be addressed to Kasper Otten.

