## [Peer Review File · Nature Communications]

Human cooperation in changing groups in a large-scale public goods gameREVIEWER COMMENTS

Reviewer #1 (Remarks to the Author):

This large-scale study is among the first to robustly investigate the impact of turn-over in group membership on public good provision. Results reveal that newcomers initially contribute comparatively less, partly because they lack the resources to do so, yet over time and with accumulated resources they contribute at par with incumbents. Findings resonate with laboratory experiments and add a 'real world' flavor to it, thus both validating experimental laboratory studies and answering questions these laboratory experiments cannot.

The paper is well-written and the material, analyses and results are properly described. There are three issues the authors could elaborate upon a bit more:

1. It is mentioned (line 138ff) that group members can sanction others (through attacks). This is not further developed but sanctions are key in developing cooperation norms. At least some more information on what these sanctions mean (are they costly to sanctioner and target?) and what form do they take (are they publicly seen by all others or not) may help to understand better the context within which current results are set.

2. Perhaps I missed it, but I couldn't find information on entry and leave decisions. What makes that a newcomer enters the group (island) -- is this at the newcomers discretion, and what information do newcomers have when deciding (not) to enter an island. Vice versa, can incumbents prevent (some) newcomers to enter. And can groups decide to exclude some members at some point. These and related questions could be covered (in the Main Text), also to better understand the turn-over dynamics observed here.

3. From the Methods and Materials, it can be seen that individuals have two public goods (woodmill across islands, and then an island-specific public good). As far as I understand, in the main analyses, contributions are taken as the sum of both general and local public goods. Have the authors insight into whether newcomers spent comparatively more to local than general public goods, and even out/reverse the longer they stay in a group? (and am I correct in my intuition that contributing to the local PG is perhaps a way to signal cooperative intention to one's island incumbents?).

Reviewer #2 (Remarks to the Author):

In this manuscript, the authors present several analyses shedding light on individuals' contributions in public goods games using an extremely large ($N \sim 135,000$, $\sim 11,300$ group, $\sim 1.5M$ decisions), longitudinal (56 weeks), and uniquely-sourced semi-field study dataset (the online game Ikariam). The sheer size of the data, as well as the ecological validity granted by such a context-rich game that closely parallels public goods games in typical behavioral designs, are sufficiently novel and compelling to warrant the paper's publication. Further, the authors provide comprehensive robustness checks on their analyses, visually and intuitively appealing figures, and clear and detailed descriptions of their unique game context. Still, I have a few major concerns regarding the authors' contextualization of the present study within the relevant literature, the clarity of their decision to discretize continuous data (or not), and the (potential) non-linearity of players' contribution incentives, as well as a few miscellaneous minor concerns. Once these are addressed, however, I believe this will be an excellent contribution to the literature and a paper of broad appeal. (I would like to add, upon having fully written out my comments, that their length is a testament to my engagement with the material – I thoroughly enjoyed reviewing this paper and look forward to its publication!)

- Major issues

- o Gaps in contextualization within the literature.

- ♣ In the Introduction, the authors are primarily concerned with motivating their investigation of changing group composition in public goods games. Yet, reading through their results, a few important concepts are introduced that prior research has examined but that the authors neglect to sufficiently motivate. I detail each of these and provide suggested references, denoting especially relevant citations with (***) below.

- ♣ First, the authors attempt to emphasize the novelty of their investigation of changing group composition in public goods games. Yet, they fail to contextualize their work within a large literature on cooperation in dynamic networks.

- In addition to reviewing the literature below, there was an important nuance in the gameplay I caught in the Methods (L.551-4) that bears on the interpretation of the authors' results and warrants discussion. Namely, the authors make it clear that while entry into groups is easy and at players' discretion, it appears that exit is quite rare unless a player quits the game. Research on cooperation in groups with changing composition (i.e. dynamic networks) typically allows for roughly equivalent ease of entry and exit, in stark contrast with the authors' setting. It would be instructive for the authors to minimally mention this discrepancy in their Discussion, and perhaps additionally speculate on its implications for cooperation.

- Evolutionary modeling

- o Cavaliere, M. et al. (2012) Prosperity is associated with instability in dynamical networks. *J. Theor. Biol.* 299, 126–138

- o Fu, F. et al. (2008) Reputation-based partner choice promotes cooperation in social networks. *Phys. Rev. E* 78, 026117

- o Perc, M. and Szolnoki, A. (2010) Coevolutionary games – a mini review. *Biosystems* 99, 109–125

- o Santos, F.C. et al. (2006) Cooperation prevails when individuals adjust their social ties. *PLoS Comput. Biol.* 2, e140

- o Skyrms, B. and Pemantle, R. (2000) A dynamic model of social network formation. *Proc. Natl. Acad. Sci. U.S.A.* 97, 9340

- o Tarnita, C.E. et al. (2009) Evolutionary dynamics in set structured populations. *Proc. Natl. Acad. Sci. U.S.A.* 106, 8601–8604

- Behavioral experiments

- o ***Rand, D.G. et al. (2011) Dynamic social networks promote cooperation in experiments with humans. *Proc. Natl. Acad. Sci. U.S.A.* 108, 19193–19198

- o Efferson, C. et al. (2008) The coevolution of cultural groups and ingroup favoritism. *Science* 321, 1844–1849

- o Fehl, K. et al. (2011) Co-evolution of behaviour and social network structure promotes human cooperation. *Ecol. Lett.* 14, 546–551

- o Jordan, J.J. et al. (2013) Contagion of cooperation in static and fluid social networks. *PLoS ONE* 8, e66199

- o Wang, J. et al. (2012) Cooperation and assortativity with dynamic partner updating. *Proc. Natl. Acad. Sci. U.S.A.* 109, 14363–14368

- ♣ The concept of social norms emerges as an important to the authors investigation (e.g. L.43, 91, 140, 415). Acknowledging the authors' citations 48 and 49, they further fail to contextualize their work within the literature building from the foundational paper below.

- ***Ohtsuki, H. and Iwasa, Y. (2006) The leading eight: social norms that can maintain cooperation by indirect reciprocity. *J. Theor. Biol.* 239, 435–444

- ♣ Although a relatively minor theme, the authors begin to discuss the role of inequality on L.362. It would be instructive if they contextualized their findings within the relevant literature on inequality in cooperation.

- ***Hauser, O. P., Hilbe, C., Chatterjee, K., & Nowak, M. A. (2019). Social dilemmas among unequals. *Nature*, 572(7770), 524-527.

- Heap, S. P. H., Ramalingam, A., & Stoddard, B. V. (2016). Endowment inequality in public goods games: A re-examination. *Economics Letters*, 146, 4-7.

- Hauser, O. P., Kraft-Todd, G. T., Rand, D. G., Nowak, M. A., & Norton, M. I. (2021). Invisible inequality leads to punishing the poor and rewarding the rich. *Behavioural Public Policy*, 5(3), 333-353.

- Martinangeli, A. F. (2021). Do what (you think) the rich will do: Inequality and belief heterogeneity in public good provision. *Journal of Economic Psychology*, 83, 102364.

- Liu, J., Peng, M., Peng, Y., Li, Y., Chu, C., Li, X., & Liu, Q. (2021). Effects of inequality on a spatial evolutionary public goods game. *The European Physical Journal B*, 94(8), 1-7.

- ♣ The authors hint at the threat of attack from other players L.137-140. Because this theme does not feature centrally to their investigation, I would encourage them to cut this text. If they do not, there is a large literature on (primarily third-party) punishment in cooperation that the authors would need to discuss. Further, the mechanism of attack, and its bearing on contributions would need to be explained in greater detail.

- o Discretizing continuous data

- ♣ I hope that this is a flaw in my interpretation rather than the authors' analyses, but given the continuous nature of the authors' data, it seems appropriate for the authors to analyze it as such. For the purpose of visualization (e.g. Figures 1 and S3), it can be useful to discretize continuous data, and I believe the authors' doing so in these cases is appropriate. However, in many of the authors' analyses, it is unclear in which manner they conducted them, and this issue must be clarified. In any cases where the issue is not merely clarity in exposition, the authors should either attempt to provide sufficiently compelling theoretical or empirical arguments to discretize their data in their analyses, or else analyze their data as continuous. I have organized my comments by variable below.

- ♣ Time period

- The authors' current use of the phrase "time period" is vague. In the legend of Figure 1, for example, "time period" is used to denote a set of time periods (e.g. 1-5, 6-10, etc). In contrast, in Figure 2 and, for example, on L.231 the authors' use of "over all 28 time periods" suggests that "time period" in these contexts refers to each point of data extraction. Minimally, the authors should employ distinct terminology for "a set of time periods" ("time period" seems appropriate here) compared to "a point of data extraction" (perhaps "round"? As in "round of data extraction"). I leave the nomenclature to the authors' discretion.

- Further, it seems most appropriate to analyze "time period" (or "round") as a continuous variable. Again, discretization for the purpose of visualization is useful, but it seems arbitrary to divide "time period" as the authors have (why not 4 rounds per period? Or 6? Etc.) without sufficiently motivated theoretical or empirical explanation.

- Because of the vagueness introduced here, I am consequently uncertain in subsequent analyses the sense in which the authors use "time period." In every case—in the text as well as in regression tables, including in the supplement—the authors should clarify this issue.

- ♣ Private good

- Related to my concern above, the authors attempt a justification of their discretization of "private good" in S5, but I find it unsatisfactory.

- I would have assumed that, parallel to typical behavioral designs, "private good" would be akin to something like "endowment." Perhaps that analogy is apt, but the explanation the authors provide (below) does not make that clear:

- o "A player's private good level on an island is captured by the player's town hall on the island. The town hall is the most important building in one's town and is also used between players to communicate their level of progress on the island."

- The justification the authors provide for their quantization (below) does not actually address the problem they express, it merely obscures it behind their quantization scheme. That is, if the problem is that the distributions of private goods for incumbents and newcomers is different, and the authors want to compare "similar private good levels" between these groups, merely quantizing by absolute levels of private goods does not achieve this goal.

- o "Had we used the private good level as a continuous variable, we would again not have had enough newcomers in the upper parts of the private good level distribution to reliably compare newcomers and incumbents with similar private good levels."

- First, I would suggest providing a visual representation of these distributions (e.g. histograms). Second, I would suggest one of two alternative analysis strategies. These data could be analyzed continuously by relative levels of private goods within group (in other words, within incumbents and newcomers, assign relative private good scores by their percentile within the distribution of absolute private goods). Alternatively, some transformation technique (e.g. log-transformation, depending on the shape of the underlying distributions) could be used to make the absolute level of private goods

more comparable.

- ♣ Number of newcomers/leavers and Group size

- Given my concerns with discretizing time period and private good, I am subsequently unsure of the authors' use of number of newcomers/leavers (especially considering Figure S3) as well as group size. The authors should clarify how these variables were used in their analyses.

- o Non-linear contribution incentives

- ♣ S2 and Table S1 gave me considerable pause, and I hope that my concern is attributable to misunderstanding rather than underexplored complexity in the game context. I ask the authors' (and editor's) forgiveness in providing some remedial exposition of typical behavioral public goods game designs first to ensure that my concern is clear.

- Public goods games have a multiplier M on individuals' contributions to the public good. Individuals' contributions are summed, multiplied by M , and distributed equally among individuals. Crucially, there is a "social dilemma" if $1 < M < N$ (where N is the number of individuals in the group) because in this range of M , contributions are costly to individuals but beneficial to the group (i.e. socially efficient). If $M < 1$, contributions are both costly to individuals and socially inefficient (thus there is no incentive to contribute) while if $M > N$, contributions are both beneficial to individuals and socially efficient (thus there is no incentive NOT to contribute). Relevant to the present concern, M is typically constant.

- Translating M to the authors' game context is non-trivial because it corresponds to an island's sawmill level, but the output of the sawmill is contingent upon time (i.e. wood per hour) rather than discrete game periods (typically, "rounds") that directly correspond to players' contribution decisions (i.e. in each round, players make contribution decisions and then the group receives the return). Thus, players do not receive a return on contributions until the sawmill meets the threshold for the next level (an issue the authors discuss), but additionally, there is presumably heterogeneity in the time islands' sawmills spend at each level. Understanding players' return on investment is further complicated by the fact that it appears that there are diminishing marginal returns to increasing saw level (although this is a point on which I am uncertain because I do not follow what the authors mean by "step return").

- Taken together, I have the impression that M effectively decreases with sawmill level, and if true, it complicates the authors' interpretation and suggests additional robustness checks. Namely, newcomers to lower sawmill level islands would have greater (prosocial, altruistic, etc.) incentive to contribute than newcomers to higher sawmill level islands. If this is the case, I would strongly suggest the authors re-run their analyses controlling for sawmill level to ensure the robustness of their results.

- Minor issues

- o I appreciate that the game context from which the authors obtain their data is novel; yet, the authors (commendably) cite two previous papers using Ikariam data (citations 39 and 40). It would be helpful—particularly to substantiate their claim of novelty—if the authors could briefly summarize how their investigation differs from these.

- o In a few places, the authors use shorthand to refer to previous arguments they have made. I appreciate that these were probably motivated by a concern for concision (and may seem clear to the authors, given their intimate knowledge of the manuscript), but I find it unclear as a reader. I would suggest that, despite the seeming repetitiveness, they elaborate anywhere this language is employed (a few examples below) to enhance readers' comprehension.

- ♣ L.274: "our second research question"

- ♣ L.334-5: "the first-mentioned theoretical mechanism"

- ♣ L.347: "the posited norm mechanism"

- o On L.420, I think the term "identity-based explanations" is misleading. The authors are commenting on whether newcomers contribute more on islands where there are players with whom they co-inhabit other islands, and this phenomenon is more aptly described by something like "familiarity" or "shared history" (or perhaps "reciprocity") rather than "identity."

- o It would be helpful for the authors to comment on the generalizability of their findings in the Discussion with regard to the representativeness of their sample. The size of their sample is, of course, remarkable, and the average age of players (31) is quite close to the global median (see citation below). The countries from which the data were collected go slightly beyond typical WEIRD samples (particularly the inclusion of Greece and Turkey; if unfamiliar, see citation below). The gender imbalance (80%) is of concern and should be addressed. It is unclear whether the authors

have access to income and/or education data (often highly correlated), which should either be included or else admitted as a limitation.

♣ Ritchie, H. and Roser, M. (2019) "Age Structure". Published online at OurWorldInData.org. Retrieved from: '<https://ourworldindata.org/age-structure>' [Online Resource]

♣ Henrich, J., Heine, S. J., & Norenzayan, A. (2010). The weirdest people in the world?. *Behavioral and brain sciences*, 33(2-3), 61-83.

o I find Figure S3 quite helpful, yet I was unable to locate the corresponding regression results for this figure. Please include these if they were omitted or clearly indicate where they can be found (in the figure legend) if they are already included.

Reviewer #3 (Remarks to the Author):

This paper investigates how changing group compositions affects contributions in a public goods game. It analyzed the data of decisions in an online game. In this game, players need resources and there are facilities that increase the efficiency of production for all players in a group. Thus, investments into these facilities can be considered as investments into public goods. The groups consist of players located on the same island. People can move to islands, but moving away is not part of the normal strategies of the game. The paper shows that incumbents contribute more than the newcomers. The analysis is very comprehensive, it includes controls of groups size, length of stay in a group, and exploits the fact that people can be in multiple groups, which even allow to control for individual characteristics.

The paper is very well written, and the analysis is comprehensive, including well-designed figures. It also links the conclusions to how migration affects contributions to the public good. The link appears convincing but it raises also some questions on what we exactly learn from the paper. The game provides a very specific environment. First, the economic environment is such that "newcomers initially have lower benefits and efficacy of contributing to public goods" (page 15, line 447f). This means that this feature of the environment could drive the result. In the conclusion it is written that this feature could also be relevant in reality. I do not disagree but the evidence in this game does not show that these incentives are also relevant in reality. It is even questionable whether it provides evidence that newcomers would provide less to a public good, in which they have equal incentives as the incumbents. This problem could be addressed with a more in depth analysis of the economic incentives in the game. Another relevant point is that people are not exogenously allocated to the groups. This means that we cannot easily make causal claims about the difference between newcomers and incumbents. On line 176, it is claimed that selection effects can be excluded because the same people can be observed as newcomers as well as incumbents. It is indeed a good feature that this allows to control for player characteristics. However, they choose to enter a new group for some reason. So, when they enter, they have a specific goal in mind. This goal can be different from the goals on the islands, in which they are incumbent. Third, that people are part of different groups has also drawbacks. Since the game runs in real time, the players have to share the attention between the different groups. I wonder about what are the best strategies how to distribute game resources and time between the different groups.

This game provides a very interesting environment that allows to investigate behavior in a very controlled but context-rich environment. However, the interpretation is not so easy because the incentives are not that easy to understand. In particular, the paper ignores incentives created by the threat of punishment and reputational incentives. It is mentioned that players can be attacked: "Community pages further warn players that they may be attacked by other players if they do not contribute their share" (line 137f). If this is a relevant mechanism then more should be known about such attacks. For example, are newcomers less attacked when they do not contribute.

Minor point

The game is very well explained in the Materials and Methods. I recommend adding more details on entry and leaving the groups in the main part of the paper.

Line 158ff, "Since group size is limited to 17 players per group, and each player can either be a newcomer or an incumbent at each biweekly measurement, we can observe up to 170 different

combinations of the number of incumbents and newcomers biweekly." Why are there 170 combinations?

Page 19, line 586, it is claimed that the game is "free of observer bias". Different to lab experiments, there is no experimenter who is visible, but there are the other player who are more visible.

Dear Reviewers,

We are very thankful for your efforts, kind remarks, and very useful comments. We have incorporated almost all of the suggestions and think the manuscript benefitted greatly from them. Below, we provide a detailed response to each comment.

Reviewer #1

Comment

This large-scale study is among the first to robustly investigate the impact of turn-over in group membership on public good provision. Results reveal that newcomers initially contribute comparatively less, partly because they lack the resources to do so, yet over time and with accumulated resources they contribute at par with incumbents. Findings resonate with laboratory experiments and add a 'real world' flavor to it, thus both validating experimental laboratory studies and answering questions these laboratory experiments cannot.

The paper is well-written and the material, analyses and results are properly described. There are three issues the authors could elaborate upon a bit more:

Response

We thank the reviewer for the kind compliments and insightful comments. We respond to each comment in detail below.

Comment

1. It is mentioned (line 138ff) that group members can sanction others (through attacks). This is not further developed but sanctions are key in developing cooperation norms. At least some more information on what these sanctions mean (are they costly to sanctioner and target?) and what form do they take (are they publicly seen by all others or not) may help to understand better the context within which current results are set.

Response

It is a valid point that sanctions can influence cooperation levels, which is especially the case when sanctions are relatively cheap to carry out for the sanctioner. In Ikariam, however, sanctions (attacks) are very costly to the sanctioner and the target, using up a lot of resources. Hence, attacks occur only infrequently and are not central to the gameplay of most players. We now describe attacks in more detail in the Materials and Methods section, lines 597-604. Attacks are not publicly seen by others and can serve multiple purposes, e.g., sanctioning but also taking away resources.

Since we do not have data on attacks, because they can serve other purposes than sanctioning, and because they are not central to gameplay, we have now moved the information on attacks to the Materials and Methods section. This way (more) information on attacks is available to the reader, but it no longer distracts from our main contributions and analyses which do not include attacks (as also requested by reviewer #2).

Comment

2. Perhaps I missed it, but I couldn't find information on entry and leave decisions. What makes that a newcomer enters the group (island) -- is this at the newcomers discretion, and what information do newcomers have when deciding (not) to enter an island. Vice versa, can incumbents prevent (some)

newcomers to enter. And can groups decide to exclude some members at some point. These and related questions could be covered (in the Main Text), also to better understand the turn-over dynamics observed here.

Response

We now explain the context of entry and leave decisions under the subsection *Setting*, lines 160-179. There, it reads:

To gain access to more resources than available in a given group, an individual can enter additional groups by building a town on additional islands. Entering additional islands is an essential part of progressing in the game, as individuals eventually will need more resources than produced on their first island(s). When players newly join an island, they are able to extract resources from the island at a rate that depends on how much the incumbents of that island have contributed so far. Entering an island thus means entering a new group with a specific state of public good provision. An individual can become part of up to 12 groups, which means that an individual can become a newcomer several times. This also means that the same individual will sometimes be an incumbent in one group and a newcomer in another.

Once an individual has entered a new group, it is generally not possible to leave, with two exceptions. The first is if an individual quits the game altogether. The second is if an individual spends real money to be able to move their town from one group to another, which is very rare. An individual can choose any group to enter, as long as the group has not reached the maximum group size of 17 yet. Incumbents thus have no say in who enters their group and cannot exclude members. When choosing which group to enter next, individuals have information on the island-specific resource that is produced, the location and size of the group, the incumbents in the group, and the current level of the public goods in the group. Generally, individuals will prefer groups that produce the island-specific resource that individuals are most in need of, groups that are located close to their current group(s), and groups with high public good levels.

We now also discuss how the entry and leave dynamics in Ikariam compare to previous research on cooperation in dynamic networks, in lines 96-111 and 463-474 (as requested by reviewer #2).

Comment

3. From the Methods and Materials, it can be seen that individuals have two public goods (woodmill across islands, and then an island-specific public good). As far as I understand, in the main analyses, contributions are taken as the sum of both general and local public goods. Have the authors insight into whether newcomers spent comparatively more to local than general public goods, and even out/reverse the longer they stay in a group? (and am I correct in my intuition that contributing to the local PG is perhaps a way to signal cooperative intention to one's island incumbents?).

Response

The reviewer is correct that we combine the contributions to the general and island-specific public goods, since there is no difference in contribution patterns. When analyzing the two public goods separately, we see that players contribute about equally to both, and the negative relationship between the number of newcomers and the contributions is (equally) present for both public goods. We show this in the supplementary material, Table S6-8, and now also mention this in the main text in the *Design* section on lines 210-216. There, it reads:

Recall that there are two public goods that individuals can contribute to in each group (the sawmill and the island-specific good). Resources that an individual contributes to one of the two public goods cannot be contributed to the other public good. The contributions to the two public goods are added up and divided by the total resources available at the time of measurement to obtain an individual's contribution percentage. Analyses that examine each public good separately are provided in the supplementary material and show no substantial differences between the two (Table S6-8).

Table S8 on the newcomer-incumbent difference in the contribution percentage per public good is newly added. This table shows that the newcomer-incumbent difference is largely similar for both public goods.

Reviewer #2

Comment

In this manuscript, the authors present several analyses shedding light on individuals' contributions in public goods games using an extremely large ($N \sim 135,000$, $\sim 11,300$ group, $\sim 1.5M$ decisions), longitudinal (56 weeks), and uniquely-sourced semi-field study dataset (the online game Ikariam). The sheer size of the data, as well as the ecological validity granted by such a context-rich game that closely parallels public goods games in typical behavioral designs, are sufficiently novel and compelling to warrant the paper's publication. Further, the authors provide comprehensive robustness checks on their analyses, visually and intuitively appealing figures, and clear and detailed descriptions of their unique game context. Still, I have a few major concerns regarding the authors' contextualization of the present study within the relevant literature, the clarity of their decision to discretize continuous data (or not), and the (potential) non-linearity of players' contribution incentives, as well as a few miscellaneous minor concerns. Once these are addressed, however, I believe this will be an excellent contribution to the literature and a paper of broad appeal. (I would like to add, upon having fully written out my comments, that their length is a testament to my engagement with the material – I thoroughly enjoyed reviewing this paper and look forward to its publication!)

Response

We are grateful for the very kind compliments, thorough engagement, and many useful comments and literature references provided by the reviewer. We respond to each comment in detail below.

Major issues

Comment

1. Gaps in contextualization within the literature.

In the Introduction, the authors are primarily concerned with motivating their investigation of changing group composition in public goods games. Yet, reading through their results, a few important concepts are introduced that prior research has examined but that the authors neglect to sufficiently motivate. I detail each of these and provide suggested references, denoting especially relevant citations with (***) below.

Response

We thank the reviewer for the useful references. We have incorporated much of the literature that the reviewer suggested and believe that this has improved our manuscript both in terms of embedding it within existing research and showing the novelty of our study. We respond to each set of literature references in more detail below.

Comment

First, the authors attempt to emphasize the novelty of their investigation of changing group composition in public goods games. Yet, they fail to contextualize their work within a large literature on cooperation in dynamic networks.

- Evolutionary modeling

- o Cavaliere, M. et al. (2012) Prosperity is associated with instability in dynamical networks. *J. Theor. Biol.* 299, 126–138

- o Fu, F. et al. (2008) Reputation-based partner choice promotes cooperation in social networks. *Phys. Rev. E* 78, 026117

- o Perc, M. and Szolnoki, A. (2010) Coevolutionary games – a mini review. *Biosystems* 99, 109–125

- o Santos, F.C. et al. (2006) Cooperation prevails when individuals adjust their social ties. *PLoS Comput. Biol.* 2, e140

- o Skyrms, B. and Pemantle, R. (2000) A dynamic model of social network formation. *Proc. Natl. Acad. Sci. U.S.A.* 97, 9340

- o Tarnita, C.E. et al. (2009) Evolutionary dynamics in set structured populations. *Proc. Natl. Acad. Sci. U.S.A.* 106, 8601–8604

- Behavioral experiments

- o ***Rand, D.G. et al. (2011) Dynamic social networks promote cooperation in experiments with humans. *Proc. Natl. Acad. Sci. U.S.A.* 108, 19193–19198

- o Efferson, C. et al. (2008) The coevolution of cultural groups and ingroup favoritism. *Science* 321, 1844–1849

- o Fehl, K. et al. (2011) Co-evolution of behaviour and social network structure promotes human cooperation. *Ecol. Lett.* 14, 546–551

- o Jordan, J.J. et al. (2013) Contagion of cooperation in static and fluid social networks. *PLoS ONE* 8, e66199

- o Wang, J. et al. (2012) Cooperation and assortativity with dynamic partner updating. *Proc. Natl. Acad. Sci. U.S.A.* 109, 14363–14368

Response

We now relate our study to the literature on cooperation in dynamic networks in the section *Backgrounds*, lines 96-111. There, it reads:

There is a related literature on cooperation in dynamic networks. In dynamic networks, actors have some control over whom they interact with, allowing them to form and break ties with others based on others' cooperation decisions. Evolutionary models show that such strategic tie formation and dissolution can promote cooperation^{39,40}. In particular, cooperation is expected to be higher if actors can frequently break with defectors and link with cooperative actors⁴¹. Behavioral experiments generally support these predictions; cooperation is higher in dynamic networks than in static networks and leads to clusters of cooperation^{42–45}. However, this literature leaves largely unaddressed what happens in situations where individuals have little say in how the composition of their group changes and hence cannot easily break with defectors. For example, residents in a neighborhood typically do not get to choose who enters or leaves and employees in work organizations frequently have to accept with whom they have to collaborate based on the decision of their employers. What is more, exit costs are typically substantial in these situations, meaning that incumbents have little option to leave if they are dissatisfied with the newcomers. In the context that we study, options to leave the group or exclude free-riders are also limited. In contrast to previous literature, this allows us to examine how group changes are related to cooperation when avoiding free-riders is not feasible.

We also come back to this literature in the Discussion, lines 463-474, where we discuss how our finding of a negative relationship between group changes and contributions differs from the typically positive association between group changes and cooperation in dynamic networks.

Comment

- The concept of social norms emerges as an important to the authors investigation (e.g. L.43, 91, 140, 415). Acknowledging the authors' citations 48 and 49, they further fail to contextualize their work within the literature building from the foundational paper below.
- ***Ohtsuki, H. and Iwasa, Y. (2006) The leading eight: social norms that can maintain cooperation by indirect reciprocity. J. Theor. Biol. 239, 435-444

Response

We have added a new paragraph to embed our study in this literature on norms and reciprocity. The paragraph, in lines 150-158, reads:

Players can see the contributions of other players on the island at any time (a screenshot is provided in supplementary material, Figure S1). This observability of the contributions of all members is an important element that allows for cooperation norms to be at play via reciprocity, where individuals can condition their contributions on their group members' contributions^{52,53}. That contributions on each island are observable to all members allows us to examine whether players indeed contribute more when their group members also contribute more. A novel contribution of our study is that it also allows us to examine to what extent a player's tendency to contribute in line with the group differs between newcomers and incumbents, and whether newcomers contribute more in line with the group as they spend more time in the group.

Adding this text also helps to clarify the reputational incentives as requested by reviewer #3. The finding that a player's own contribution relates positively to the group members' contributions, and that this is initially less so for newcomers but increases with their time spent in the group is also related to norms on lines 367-382.

Comment

- Although a relatively minor theme, the authors begin to discuss the role of inequality on L.362. It would be instructive if they contextualized their findings within the relevant literature on inequality in cooperation.
- ***Hauser, O. P., Hilbe, C., Chatterjee, K., & Nowak, M. A. (2019). Social dilemmas among unequals. Nature, 572(7770), 524-527.
- Heap, S. P. H., Ramalingam, A., & Stoddard, B. V. (2016). Endowment inequality in public goods games: A re-examination. Economics Letters, 146, 4-7.
- Hauser, O. P., Kraft-Todd, G. T., Rand, D. G., Nowak, M. A., & Norton, M. I. (2021). Invisible inequality leads to punishing the poor and rewarding the rich. Behavioural Public Policy, 5(3), 333-353.
- Martinangeli, A. F. (2021). Do what (you think) the rich will do: Inequality and belief heterogeneity in public good provision. Journal of Economic Psychology, 83, 102364.
- Liu, J., Peng, M., Peng, Y., Li, Y., Chu, C., Li, X., & Liu, Q. (2021). Effects of inequality on a spatial evolutionary public goods game. The European Physical Journal B, 94(8), 1-7.

Response

In the discussion section, on lines 496-502, we now relate our findings to prior literature on inequality and cooperation. In particular, we suggest how our finding on inequality – that the newcomer-incumbent difference in contributions is associated with the newcomer-incumbent difference in benefits/efficacy of contributing – is in line with prior theory and some empirical research suggesting that inequality can hamper cooperation.

Comment

In addition to reviewing the literature, there was an important nuance in the gameplay I caught in the Methods (L.551-4) that bears on the interpretation of the authors' results and warrants discussion. Namely, the authors make it clear that while entry into groups is easy and at players' discretion, it appears that exit is quite rare unless a player quits the game. Research on cooperation in groups with changing composition (i.e. dynamic networks) typically allows for roughly equivalent ease of entry and exit, in stark contrast with the authors' setting. It would be instructive for the authors to minimally mention this discrepancy in their Discussion, and perhaps additionally speculate on its implications for cooperation.

Response

We have taken several steps to clarify the difference in the entry and leave mechanics between Ikariam and other literature on dynamic networks and the potential implications. The difference is first announced in the newly added section on dynamic networks (lines 96-111). In addition, the information on the entry and leave mechanics of Ikariam is now moved to the main text, in the section Settings (lines 160-179) and the text has been extended. Finally, the Discussion section (lines 463-474) describes how our findings differ from other findings in the dynamic networks literature and how this might be related to the difficulty of avoiding or breaking ties with free-riding individuals in Ikariam.

Comment

The authors hint at the threat of attack from other players L.137-140. Because this theme does not feature centrally to their investigation, I would encourage them to cut this text. If they do not, there is a large literature on (primarily third-party) punishment in cooperation that the authors would need to discuss. Further, the mechanism of attack, and its bearing on contributions would need to be explained in greater detail.

Response

We agree with the reviewer that attacks should feature less centrally in our manuscript as they are not part of the analysis. Therefore, we have now moved the information on attacks to the Materials and Methods section (lines 597-604). This way the information on attacks no longer distracts from our main contributions and analyses, but it is still available to provide the full context of the game. Please see also our response to comment 1 of reviewer 1.

Comment

2. Discretizing continuous data

I hope that this is a flaw in my interpretation rather than the authors' analyses, but given the continuous nature of the authors' data, it seems appropriate for the authors to analyze it as such. For the purpose of visualization (e.g. Figures 1 and S3), it can be useful to discretize continuous data, and I believe the authors' doing so in these cases is appropriate. However, in many of the authors' analyses, it is unclear in which manner they conducted them, and this issue must be clarified. In any cases where the issue is not merely clarity in exposition, the authors should either attempt to provide

sufficiently compelling theoretical or empirical arguments to discretize their data in their analyses, or else analyze their data as continuous. I have organized my comments by variable below.

Response

Discretizing data was indeed mainly done for the purpose of visualization. We agree that we should make this clearer and have taken several steps to clarify that we analyze the data without discretizing (and changed the few instances in which we previously did analyze discretized data). We specify the details per reviewer comment and variable below.

Comment

Time period

- The authors' current use of the phrase "time period" is vague. In the legend of Figure 1, for example, "time period" is used to denote a set of time periods (e.g. 1-5, 6-10, etc). In contrast, in Figure 2 and, for example, on L.231 the authors' use of "over all 28 time periods" suggests that "time period" in these contexts refers to each point of data extraction. Minimally, the authors should employ distinct terminology for "a set of time periods" ("time period" seems appropriate here) compared to "a point of data extraction" (perhaps "round"? As in "round of data extraction"). I leave the nomenclature to the authors' discretion.
- Further, it seems most appropriate to analyze "time period" (or "round") as a continuous variable. Again, discretization for the purpose of visualization is useful, but it seems arbitrary to divide "time period" as the authors have (why not 4 rounds per period? Or 6? Etc.) without sufficiently motivated theoretical or empirical explanation.
- Because of the vagueness introduced here, I am consequently uncertain in subsequent analyses the sense in which the authors use "time period." In every case—in the text as well as in regression tables, including in the supplement—the authors should clarify this issue.

Response

We agree that the use of time period when referring to multiple time periods is confusing in Figure 1. We now more clearly specify how we define a time period on lines 182-185: "The data are systematically structured in 28 biweekly intervals, i.e., we have one observation per two weeks per player-group combination. We refer to each biweekly interval as a time period, so we have 28 time periods.". We do not show data points for each separate period in Figure 1 as that would make the figure too crowded for visualization purposes. To make clearer that each data point of the figure refers to multiple periods, the figure legend now refers to periods as plural. That is, instead of referring to period 1-5, period 6-10, etc., we now refer to periods 1-5, periods 6-10, and so on. Moreover, in the caption for Figure 1, we have now added the text "The data are discretized in this figure for visualization purposes, the non-discretized analyses can be found in Table 1.". In the main text between describing Figure 1 and reporting the statistical analyses, we write "Although the variables are discretized in Figure 1 for visualization purposes, we do not discretize data in any of the statistical analyses and we find the same patterns there." (lines 246-247).

The only other instance in which we discretized the time period variable was in the model for Figure S3, again for visualization purposes. We now also include a statement in the figure caption there "The data are discretized in this figure and its underlying model (Table S14) for visualization purposes, the non-discretized analyses can be found in Table S15". In all other instances, the time period variable and the other variables were indeed analyzed without discretizing, and we now clarify this in the table notes for all analyses.

Comment

Private good

- Related to my concern above, the authors attempt a justification of their discretization of “private good” in S5, but I find it unsatisfactory.
- I would have assumed that, parallel to typical behavioral designs, “private good” would be akin to something like “endowment.” Perhaps that analogy is apt, but the explanation the authors provide (below) does not make that clear:
 - “A player’s private good level on an island is captured by the player’s town hall on the island. The town hall is the most important building in one’s town and is also used between players to communicate their level of progress on the island.”
- The justification the authors provide for their quantization (below) does not actually address the problem they express, it merely obscures it behind their quantization scheme. That is, if the problem is that the distributions of private goods for incumbents and newcomers is different, and the authors want to compare “similar private good levels” between these groups, merely quantizing by absolute levels of private goods does not achieve this goal.
 - “Had we used the private good level as a continuous variable, we would again not have had enough newcomers in the upper parts of the private good level distribution to reliably compare newcomers and incumbents with similar private good levels.”
- First, I would suggest providing a visual representation of these distributions (e.g. histograms). Second, I would suggest one of two alternative analysis strategies. These data could be analyzed continuously by relative levels of private goods within group (in other words, within incumbents and newcomers, assign relative private good scores by their percentile within the distribution of absolute private goods). Alternatively, some transformation technique (e.g. log-transformation, depending on the shape of the underlying distributions) could be used to make the absolute level of private goods more comparable.

Response

We thank the reviewer for expressing these concerns. We have taken several steps to address them.

First, we now clarify how a player’s town hall level (what we call the private good level) influences the player’s resource amount and hence is akin to the endowment in public good games in S5 of the supplementary material on page 20. We write:

A player’s private good level on an island is captured by the player’s town hall level on the island. Every increase in the town hall level increases the maximum number of citizens allowed in one’s town. Each citizen works to produce resources and pays taxes to give gold (the latter of which can be used to buy yet more resources). Hence, a higher level town hall means more resources. In this sense, the town hall level is akin to the endowment in regular public good games.

The private good level is now also explained in the main text on lines 407-412.

Second, we now provide the histograms of the private good levels (town hall levels) for both incumbents and newcomers as requested, in supplementary material Figure S4. The figure shows the large difference in the distributions between newcomers and incumbents, with newcomers having considerably lower private good levels.

Third, we have replaced the discretization with continuous analyses. We agree that discretizing the town hall level is a suboptimal method to compare newcomers and incumbents with similar town hall

levels. However, we fear that the two methods suggested by the reviewer also do not fully solve the issue. When comparing newcomers and incumbents with equal percentiles (or some other transformation of the town hall level), large differences in town hall levels remain. For instance, when comparing newcomers that score above the 90th percentile of the newcomer distribution in town hall levels with incumbents that rank above the 90th percentile of the incumbent distribution in town hall levels, we are comparing newcomers with an average town hall level of ~7 with incumbents with an average town hall level of ~30 (see also the two distributions in Figure S4).

Our approach now is to control for town hall level as a *continuous* variable while including in this analysis a range of town hall levels where we have enough observations from both newcomers and incumbents reaching these town hall levels. The histogram (Figure S4) informs this range of town hall levels. We see that town hall levels above 10 are achieved by only 1% of the newcomers, making it difficult to find enough newcomers above this level to compare with incumbents of the same town hall level. Therefore, we set the range to include town hall levels from 1 to 10, which captures 99% of the newcomers and 50% of the incumbents. In Table S20, we now show that the newcomer-incumbent difference in the contribution percentage is 9% when controlling for town hall level as a continuous variable this way. The previous discretized analysis suggested a newcomer-incumbent difference in contribution percentage of 10%, so the results from the new continuous analysis are largely similar to the previous (and now removed) discretized analysis.

Comment

Number of newcomers/leavers and Group size

- Given my concerns with discretizing time period and private good, I am subsequently unsure of the authors' use of number of newcomers/leavers (especially considering Figure S3) as well as group size. The authors should clarify how these variables were used in their analyses.

Response

The number of newcomers/leavers and group size are analyzed continuously, and we have now clarified this in the table notes of the analyses. The only exception is the analysis underlying Figure S3 in the supplementary material, where the variables were entered discretized for visualization purposes. This is now explained in the legend of this figure. We have now also included a continuous counterpart to the analysis of this figure, which is provided in Table S15.

Comment

3. Non-linear contribution incentives

S2 and Table S1 gave me considerable pause, and I hope that my concern is attributable to misunderstanding rather than underexplored complexity in the game context. I ask the authors' (and editor's) forgiveness in providing some remedial exposition of typical behavioral public goods game designs first to ensure that my concern is clear.

- Public goods games have a multiplier M on individuals' contributions to the public good. Individuals' contributions are summed, multiplied by M , and distributed equally among individuals. Crucially, there is a "social dilemma" if $1 < M < N$ (where N is the number of individuals in the group) because in this range of M , contributions are costly to individuals but beneficial to the group (i.e. socially efficient). If $M < 1$, contributions are both costly to individuals and socially inefficient (thus there is no incentive to contribute) while if $M > N$, contributions are both beneficial to individuals and socially efficient (thus there is no incentive NOT to contribute). Relevant to the present concern, M is typically constant.

- Translating M to the authors' game context is non-trivial because it corresponds to an island's sawmill level, but the output of the sawmill is contingent upon time (i.e. wood per hour) rather than discrete game periods (typically, "rounds") that directly correspond to players' contribution decisions (i.e. in each round, players make contribution decisions and then the group receives the return). Thus, players do not receive a return on contributions until the sawmill meets the threshold for the next level (an issue the authors discuss), but additionally, there is presumably heterogeneity in the time islands' sawmills spend at each level. Understanding players' return on investment is further complicated by the fact that it appears that there are diminishing marginal returns to increasing saw level (although this is a point on which I am uncertain because I do not follow what the authors mean by "step return").
- Taken together, I have the impression that M effectively decreases with sawmill level, and if true, it complicates the authors' interpretation and suggests additional robustness checks. Namely, newcomers to lower sawmill level islands would have greater (prosocial, altruistic, etc.) incentive to contribute than newcomers to higher sawmill level islands. If this is the case, I would strongly suggest the authors re-run their analyses controlling for sawmill level to ensure the robustness of their results.

Response

We have now clarified the concept of step returns in section S2 of the supplementary material and how a similar concept can be used in Ikariam:

One important determinant of contributions to public goods is the value of the public good relative to the costs of producing it. For continuous public good games, this tradeoff between value and costs has been formalized in the multiplication factor and the marginal per capita return (MPCR). The parallel concept in threshold public good games is the step return, which is the total group payoff from the public good divided by the total contribution threshold⁴.

In Ikariam, the total group payoff is mostly a function of time (and group size) because increasing the public good level increases the hourly production of resources. Since the multiplication factor and MPCR are about the tradeoff between value and costs, we denote the step return in Ikariam as the inverse of the number of days until the increase in total group payoffs from levelling up the public good matches the total contribution threshold. In Table S2, we provide the step returns for a medium-sized group (8 members) associated with each public good level. A step return of 0.2 would mean that it takes a medium-sized group 5 days until the increase in the total group payoffs from levelling up the public good breaks even with the contributions that were required to surpass the threshold.

It is indeed true that the step return generally decreases with higher public good levels, meaning that there are stronger incentives to contribute to the public good at lower public good levels. We appreciate the reviewer's suggestion for the robustness analysis that controls for the public good level. We have now included this robustness analysis in the supplementary material Table S13 and S21 and also refer to it in the main text in lines 300-301 and 433-438. Table S13 shows that the negative relationship between the number of newcomers and the contribution percentage remains significant (but lowers somewhat in magnitude) after controlling for the public good level. This table also shows that the relationship becomes more negative with higher public good levels. Table S21 shows that the difference in the contribution percentage between newcomers and incumbents remains significant when controlling for the public good level, and that the difference is lowest for low public good levels.

Minor issues

Comment

I appreciate that the game context from which the authors obtain their data is novel; yet, the authors (commendably) cite two previous papers using Ikariam data (citations 39 and 40). It would be helpful—particularly to substantiate their claim of novelty—if the authors could briefly summarize how their investigation differs from these.

Response

We now shortly describe how the current study differs from the previous two Ikariam studies on lines 140-144:

(...) two prior studies have shown that Ikariam players use contribution strategies that can be categorized as free-riding, conditional cooperation, and high cooperation^{46,47}, which are commonly found in lab experiments with public goods games^{48,49}. However, these studies did not examine how group changes relate to cooperation and whether newcomers and incumbents contribute differently.

Comment

In a few places, the authors use shorthand to refer to previous arguments they have made. I appreciate that these were probably motivated by a concern for concision (and may seem clear to the authors, given their intimate knowledge of the manuscript), but I find it unclear as a reader. I would suggest that, despite the seeming repetitiveness, they elaborate anywhere this language is employed (a few examples below) to enhance readers' comprehension.

☐ L.274: “our second research question”

☐ L.334-5: “the first-mentioned theoretical mechanism”

☐ L.347: “the posited norm mechanism”

Response

We have now removed the shorthand and instead repeat the research questions/mechanisms in such cases (lines 304-306, 367-370, and 380-382).

Comment

On L.420, I think the term “identity-based explanations” is misleading. The authors are commenting on whether newcomers contribute more on islands where there are players with whom they co-inhabit other islands, and this phenomenon is more aptly described by something like “familiarity” or “shared history” (or perhaps “reciprocity”) rather than “identity.”

Response

We agree with the reviewer and now describe this phenomenon as shared history (lines 456-458).

Comment

It would be helpful for the authors to comment on the generalizability of their findings in the Discussion with regard to the representativeness of their sample. The size of their sample is, of course, remarkable, and the average age of players (31) is quite close to the global median (see citation below). The countries from which the data were collected go slightly beyond typical WEIRD samples (particularly the inclusion of Greece and Turkey; if unfamiliar, see citation below). The gender imbalance (80%) is of concern and should be addressed. It is unclear whether the authors have access

to income and/or education data (often highly correlated), which should either be included or else admitted as a limitation.

☒ Ritchie, H. and Roser, M. (2019) "Age Structure". Published online at OurWorldInData.org. Retrieved from: '<https://ourworldindata.org/age-structure>' [Online Resource]

☒ Henrich, J., Heine, S. J., & Norenzayan, A. (2010). The weirdest people in the world?. *Behavioral and brain sciences*, 33(2-3), 61-83.

Response

We thank the reviewer for the suggestion and data, which we have included in a new paragraph of the Discussion where we describe the generalizability of our findings (lines 522-533):

Compared to typical research using public good games, our sample is broader and more heterogeneous. The inclusion of players from Germany, the United Kingdom, France, Greece, and Turkey means our sample goes slightly beyond typical WEIRD samples (Western, Educated, Industrialized, Rich, and Democratic)⁶⁹. A survey reporting the average age of Ikariam players to be around 31 years⁴⁶ suggests that our sample more closely reflects the global median age than most other public good game studies which predominantly recruit younger university undergraduates⁷⁰. Whereas social dilemma studies are typically somewhat overrepresented by women⁷, Ikariam is largely overrepresented by men (~80% men) as is common for computer games. We do not have access to data on the income or education of Ikariam players, so cannot establish representativeness in these aspects. Altogether, our sample presents an improvement in terms of representativeness in some areas (e.g., global coverage and age), but still has limited representativeness in other areas (e.g., sex).

Comment

I find Figure S3 quite helpful, yet I was unable to locate the corresponding regression results for this figure. Please include these if they were omitted or clearly indicate where they can be found (in the figure legend) if they are already included.

Response

The regression results are now included in Table S14 and referenced in the figure legend. As mentioned in response to an earlier comment from the reviewer, this is one of the few cases in which we did discretize data for visualization purposes. We now also include comparable regression results in which the number of newcomers and leavers are treated as continuous variables in Table S15.

Reviewer #3

Comment

This paper investigates how changing group compositions affects contributions in a public goods game. It analyzed the data of decisions in an online game. In this game, players need resources and there are facilities that increase the efficiency of production for all players in a group. Thus, investments into these facilities can be considered as investments into public goods. The groups consist of players located on the same island. People can move to islands, but moving away is not part of the normal strategies of the game. The paper shows that incumbents contribute more than the newcomers. The analysis is very comprehensive, it includes controls of groups size, length of stay in a group, and exploits the fact that people can be in multiple groups, which even allow to control for individual characteristics.

Response

We thank the reviewer for the useful and insightful comments. We reply to each comment in detail below.

Comment

The paper is very well written, and the analysis is comprehensive, including well-designed figures. It also links the conclusions to how migration affects contributions to the public good. The link appears convincing but it raises also some questions on what we exactly learn from the paper. The game provides a very specific environment. First, the economic environment is such that “newcomers initially have lower benefits and efficacy of contributing to public goods” (page 15, line 447f). This means that this feature of the environment could drive the result. In the conclusion it is written that this feature could also be relevant in reality. I do not disagree but the evidence in this game does not show that these incentives are also relevant in reality. It is even questionable whether it provides evidence that newcomers would provide less to a public good, in which they have equal incentives as the incumbents. This problem could be addressed with a more in depth analysis of the economic incentives in the game.

Response

We thank the reviewer for the kind compliments. To address the reviewer’s point on incentives, we have now described the incentives in the Ikariam game in more depth in three main aspects:

1. Players that enter a new group by building a new town on a new island indeed have lower incentives to contribute. Players that build a new town start with a town hall level of 1, which generally means they have few resources available. This is the case because every increase in the town hall level increases the maximum number of citizens allowed in one’s town. And each citizen works to produce resources and pays taxes to give gold (the latter of which can be used to buy yet more resources). Hence, a higher level town hall means more citizens and therefore more resources. In this sense, the town hall level is akin to the endowment in regular public good games. Because newcomers generally start with lower town hall levels than incumbents (i.e., they have fewer resources), they may have an incentive to contribute less than incumbents. Indeed, we find that controlling for the town hall level decreases the newcomer-incumbent difference in the contribution percentage. However, a sizeable difference does remain even after controlling for town hall level, which suggests that newcomers contribute still less than incumbents also when their incentives are more similar to incumbents. Whereas the incentives of the town hall level and its implication for the newcomer-incumbent difference were discussed relatively briefly in the previous version of the manuscript, we now discuss them in more detail in lines 407-421 of the main text and page 20 of the supplementary material.

2. Incentives to contribute further depend on the public good level. At lower public good levels, the value of upgrading the public good relative to the costs of doing so is relatively more profitable than upgrading already high public good levels. This means that the economic incentives to contribute are generally higher when public good levels are low. In this situation, both newcomers and incumbents are in a position to effectively contribute to the public good and benefit from doing so. In contrast, if public good levels are high, it takes more resources to increase the public good level and returns are lower. In this situation, incumbents are in a better position than newcomers to effectively contribute due to their higher amount of resources. Accordingly, we find that the newcomer-incumbent difference in the contribution percentage is larger at higher public good levels. However, even at low public good levels, we still find a newcomer-incumbent difference, which again suggests that this difference holds across a wide range of incentives. We now discuss the incentives related to the public

good level and its implications for the newcomer-incumbent difference in contributions in more detail in lines 423-438 of the main text (and supplementary material section S2 and Table S13 and Table S21), which also addresses a point made by reviewer 2 on the incentives to contribute to the public goods (on page 9-10 of this response letter).

3. Players can observe the contributions of all their group members at all times in Ikariam. This means that players who free-ride will be easily detected. This observability of the contributions also allows for reciprocity incentives, where persons can condition their contributions on their group members' contributions. We discuss the incentives regarding the observability of contributions and reciprocity now in more detail in a newly added paragraph on lines 150-158, also in response to a related comment of reviewer 2 (page 5 of this letter).

Finally, although each social environment has its own particular features and incentives, Ikariam included, we note that our findings resonate with several findings on real-world newcomer-incumbent differences, which we describe in the Discussion. For instance, our finding that newcomers initially contribute less than incumbents but increase their contribution over time is in line with studies suggesting that residents are more likely to volunteer at community events if they are longer part of the community⁵⁸, that workers' output in organizations is higher with higher tenure⁵⁹, and that immigrants contribute more to charitable organizations with more time spent in the country⁶⁰.

Comment

Another relevant point is that people are not exogenously allocated to the groups. This means that we cannot easily make causal claims about the difference between newcomers and incumbents. On line 176, it is claimed that selection effects can be excluded because the same people can be observed as newcomers as well as incumbents. It is indeed a good feature that this allows to control for player characteristics. However, they choose to enter a new group for some reason. So, when they enter, they have a specific goal in mind. This goal can be different from the goals on the islands, in which they are incumbent.

Response

It is true that newcomers are not exogenously allocated to groups in Ikariam. That newcomers have some say in which groups they enter is a common issue in observational research looking at newcomer-incumbent differences, e.g., in research on differences between immigrants and natives, new and old residents in neighborhoods, and new and old employees in work organizations. In the Discussion, lines 476-494, we note that most previous observational studies could typically only observe individuals at one point in time, so changes over time within individuals as they switch roles are not accounted for. In our study, the same individuals take both the roles of newcomer and incumbent, allowing us to compare within individuals how contribution behavior changes depending on one's role in the group, as the reviewer also notes. In this respect, we think our study does provide an improvement over related research on newcomer-incumbent differences outside of the laboratory.

We now also more clearly highlight on lines 299-300 and 313-315 that we also ran crossed fixed effects models that allow to simultaneously control for player and group characteristics. Previously we only included a crossed fixed effects model for the relationship between the number of newcomers and the contribution percentage. Now we also included a crossed fixed effects model for the newcomer-incumbent difference in the contribution percentage. This model is reported in Table S12 and shows that newcomers' contribution percentage is lower than that of incumbents when controlling for both player and group characteristics.

Furthermore, the goal of both incumbents and newcomers is generally the same in Ikariam (also across different islands), i.e., to advance their town(s). Hence, differences in goals should not play a role in Ikariam. To assess this empirically, we now ran additional analyses in which we select only players who inhabit one island, i.e., players who belong to only one group. In these cases, the issue of having separate goals for separate islands is not present, and we still find a clear newcomer-incumbent difference (Table S18). These new analyses are also a response to the next comment of the reviewer.

Finally, we believe that lab experiments are certainly useful for studying newcomer-incumbent differences as they are ideal for causal inference. We now note explicitly in the discussion section on lines 539-541 that experimental research is desirable for causal inference. Although these experiments will generally not be able to track individuals over several months as our data allows, it is the combination of different data sources and methods that together provide a fuller understanding of cooperation dynamics in changing groups.

Comment

Third, that people are part of different groups has also drawbacks. Since the game runs in real time, the players have to share the attention between the different groups. I wonder about what are the best strategies how to distribute game resources and time between the different groups.

Response

Switching between groups is just a matter of one click in an easily accessible menu. Hence, time and effort is directed to the imminent goals that each player has at a given moment, i.e. collect more resources for advancing their town(s). Thus, attention sharing is not an issue. To further address the concern that players have to share their attention between different groups, we have included a new analysis that selects only players that inhabit one island, i.e., players who are part of only one group. The analysis is provided in Table S18 and referred to in the main text on lines 389-392. It shows that the newcomer-incumbent difference is also present and large when selecting only players that are part of one group and hence do not have to share their attention between different groups.

Comment

This game provides a very interesting environment that allows to investigate behavior in a very controlled but context-rich environment. However, the interpretation is in not so easy because the incentives are not that easy to understand. In particular, the paper ignores incentives created by the treat of punishment and reputational incentives. It is mentioned that players can be attacked: "Community pages further warn players that they may be attacked by other players if they do not contribute their share" (line 137f). If this is a relevant mechanism than more should be known about such attacks. For example, are newcomer less attacked when they do not contribute.

Response

We have provided more information on the incentives with regards to the observability of contributions, the town hall level (private good level), and the public good level. These changes are outlined in our response to comment 1 of this reviewer.

Attacks happen only infrequently, they are not a central mechanism to group changes or contributions. Therefore, we have decided to move the information on attacks to the Materials and Methods section (lines 597-604). In that section, we describe that attacks can serve multiple purposes, one of which is indeed sanctioning deviations from cooperation norms. However, another main purpose of attacks is to take resources away from other players. By moving the information on attacks

to the Materials and Methods section, the information is still available to the reader, but it no longer distracts from our main contributions and analyses which do not include attacks (as also requested by reviewer #2).

Minor points

Comment

The game is very well explained in the Materials and Methods. I recommend adding more details an entry and leaving the groups in the main part of the paper.

Response

We have moved the details on the entry and leave dynamics from the Materials and Methods to the main text as requested. They are now discussed in the Introduction, section Setting, lines 160-179.

Comment

Line 158ff, "Since group size is limited to 17 players per group, and each player can either be a newcomer or an incumbent at each biweekly measurement, we can observe up to 170 different combinations of the number of incumbents and newcomers biweekly." Why are there 170 combinations?

Response

The number of possible group configurations is always the group size + 1. For example, if the group size is 1 there are two possibilities: 1 newcomer or 1 incumbent; if the group size is 2 there are three possibilities: 2 newcomers, 2 incumbents, or 1 newcomer and 1 incumbent, and so on. Adding up all the possible group configurations, starting with a group size of 1 and ending with a group size of 17, gives: $2+3+4+5+6+7+8+9+10+11+12+13+14+15+16+17+18=170$. Hence, there are 170 possible different combinations of the number of incumbents and newcomers.

To avoid confusion, we now no longer specify the precise number of 170 but instead formulate the sentence as follows (lines 189-191): *Since group size ranges from 1 to 17 players per group, and each player can either be a newcomer or an incumbent at each time period, we can observe many different combinations of the number of incumbents and newcomers.*

Comment

Page 19, line 586, it is claimed that the game is "free of observer bias". Different to lab experiments, there is no experimenter who is visible, but there are the other player who are more visible.

Response

We now clarify this in the text as follows: *free of observer bias from experimenters' presence* (line 632). We do think that a game context played privately at home is a real improvement in terms of reducing observer bias over public good games in laboratory contexts (where players often also get to see other players' contributions).

REVIEWER COMMENTS

Reviewer #1 (Remarks to the Author):

The revised manuscript satisfactorily dealt with my earlier comments and suggestions. I have no further issues and thank the authors for their responsiveness.

Reviewer #2 (Remarks to the Author):

The authors have endeavored a truly superlative revision on their already impressive manuscript. The thoroughness of their responses was exemplary, as they provided detailed and careful consideration of every minor reviewer concern. Further, they have gone above and beyond in their integration of the literature I suggested, providing thought-provoking discussion of their findings in light of this work that surpassed my own consideration of these connections. I am confident that the current, strengthened manuscript will impress and inspire readers, and commend the authors for their diligent effort in such a massive undertaking. I look forward to seeing the paper in print!

Reviewer #3 (Remarks to the Author):

This revision addresses all the concerns with the previous submission. There is a small issue mentioned below. But apart from this, I consider the paper as publishable.

The text referring to causality in the conclusion is good. However, the statement on line 206f emphasizing the that selection effects can be excluded should be adjusted. I recommend writing “we can exclude selection effects based on different personal characteristics” instead of “we can exclude selection effects“. The reason is that it cannot be excluded that becoming a newcomer is related to certain (personal and game related) situations.

Reviewers' comments

Reviewer #1

Comment

The revised manuscript satisfactorily dealt with my earlier comments and suggestions. I have no further issues and thank the authors for their responsiveness.

Response

We thank the reviewer for the help with improving the manuscript!

Reviewer #2

Comment

The authors have endeavored a truly superlative revision on their already impressive manuscript. The thoroughness of their responses was exemplary, as they provided detailed and careful consideration of every minor reviewer concern. Further, they have gone above and beyond in their integration of the literature I suggested, providing thought-provoking discussion of their findings in light of this work that surpassed my own consideration of these connections. I am confident that the current, strengthened manuscript will impress and inspire readers, and commend the authors for their diligent effort in such a massive undertaking. I look forward to seeing the paper in print!

Response

We are very thankful to the reviewer for the help with improving the manuscript and the very kind remarks!

Reviewer #3

Comment

This revision addresses all the concerns with the previous submission. There is a small issue mentioned below. But apart from this, I consider the paper as publishable.

The text referring to causality in the conclusion is good. However, the statement on line 206f emphasizing the that selection effects can be excluded should be adjusted. I recommend writing "we can exclude selection effects based on different personal characteristics" instead of "we can exclude selection effects". The reason is that it cannot be excluded that becoming a newcomer is related to certain (personal and game related) situations.

Response

We thank the reviewer for the help with improving the manuscript and for the text suggestion, which we have incorporated in the revised manuscript.